# What are Key Factors for Updates in RL for LLM Reasoning?

## Abstract

Reinforcement Learning from Verifiable Rewards (RLVR) has emerged as a promising framework for enhancing the reasoning ability of large language models. However, much of the existing work is guided by heuristic intuition, leading to divergent algorithmic choices, even contradictory ones that nevertheless report empirical gains. To better understand this phenomenon, we conduct a theoretical analysis of RLVR updates. Our study reveals that differences in off-policy degree, determined by the number of gradient steps per rollout, substantially affect the distribution of importance sampling ratios and their clipping behavior, thereby altering which tokens dominate the update. Building on this insight, we characterize gradient expectation as the central quantity governing update dynamics and analyze the roles of token probability, advantage, and importance sampling ratio. Motivated by these findings, we propose Adaptive Clip Policy Optimization (ACPO), which adjusts clipping boundaries across token groups according to the empirical variance of their importance sampling ratios. Experiments on 3B and 7B models across diverse reasoning benchmarks, spanning mathematical problem solving, tabular QA, and logic puzzles, demonstrate that ACPO outperforms strong baselines such as DAPO and CISPO. These results demonstrate that principled, analysis-driven approaches yield more robust and effective RLVR methods. Code is available in: https://anonymous.4open.science/r/ACPO

## 1. Introduction

Reinforcement Learning from Verifiable Rewards (RLVR) aims to enhance the reasoning ability of large language models (LLMs) by optimizing against verifiable outcomes, such as mathematical correctness or logical validity (Shao et al., 2024; Guo et al., 2025a; Lambert et al., 2024; Jaech et al., 2024; Chen et al., 2025c; Yang et al., 2025a). This setting provides a scalable way to directly reward structured reasoning, making RLVR a central approach for advancing reliability in complex problem-solving tasks.

Despite substantial progress, RLVR research remains largely driven by heuristics rather than systematic analysis. Accordingly, many algorithmic variants have been proposed (Yu et al., 2025; Yue et al., 2025; Su et al., 2025; Zheng et al., 2025a), yet their effectiveness is still not well understood. As illustrated in Figure 1(a), even seemingly contradictory token-level heuristics report similar gains: Wang et al. (2025b) prioritize high-entropy tokens and mask low-entropy ones, arguing that uncertain tokens are pivotal for reasoning, whereas Yang et al. (2025b) emphasize high-probability tokens to prevent low-probability tokens from dominating gradient updates. Since token entropy is inversely correlated with token probability, these prescriptions are effectively opposite, yet both work in practice.

Motivated by the importance of token-level understanding in RLVR (Yang et al., 2025b; Wang et al., 2025b; Cui et al., 2025; Chen et al., 2025a; Wang et al., 2025a), we study the learning dynamics of clipped policy-gradient updates in RLVR. Our starting point is a simple but under-emphasized observation: seemingly contradictory token-selection heuristics are often evaluated under *different degrees of off-policy*. For instance, the high-entropy update strategy uses many more updates per rollout than the high-probability emphasis setting, causing the optimized policy to drift further from the behavior policy that generated the rollouts. In practice, such off-policy regimes can be unavoidable because updates are often the bottleneck: one can cheaply generate many rollouts, but can only fit a limited number of update samples per step, making it natural to reuse each rollout across multiple updates.

Because GRPO (Shao et al., 2024) relies on a clipped importance-sampling (IS) ratio, off-policy degree affects both token-gradient magnitude and *which tokens survive clipping and contribute updates*. This implies that analyses based on per-token intuition can be misleading: the overall update is an aggregation over tokens, and clipping can

[1]Anonymous Institution, Anonymous City, Anonymous Region, Anonymous Country. Correspondence to: Anonymous Author <anon.email@domain.com>.

Preliminary work. Under review by the International Conference on Machine Learning (ICML). Do not distribute.

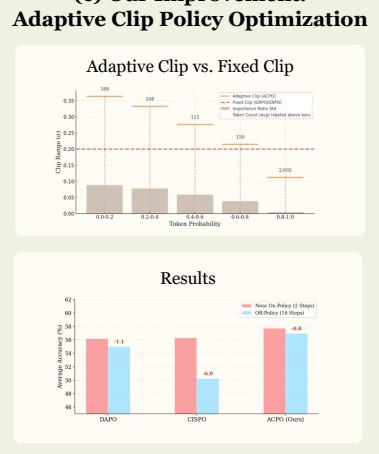

**(a) Seemingly Conflicting Results**

Updating only **high-entropy** tokens: "Forking" tokens **improves** performance

Updating only **low-probability** tokens: Over-dominance **harms** performance

**(b) Unified Analysis for Gradient Dynamics**

Key Factors Controlling Gradients:
Token Probability ($\pi$)

Advantage (A)        Importance Sampling($\rho$)

**Conflicts stem from Off-Policy Degree!**

Low-probability tokens dominate (small $\bar{\sigma}_\rho^2$)

High-probability tokens dominate (large $\bar{\sigma}_\rho^2$)

On-Policy        Off-Policy

Off-Policy Degree ($\bar{\sigma}_\rho^2$)

**(c) Our Improvement: Adaptive Clip Policy Optimization**

Adaptive Clip vs. Fixed Clip

Results

*Figure 1.* (a) Conflicting results: high-entropy updates help, while low-probability dominance hurts stability. (b) Our analysis attributes this to the clipped update, which shifts with off-policy degree via token probability, advantage, and IS. (c) ACPO adapts clipping per token group, improving over DAPO/CISPO across three reasoning tasks and off-policy regimes.

systematically suppress different token groups, changing both the update magnitude and direction. Instead, one must examine the **expected effective gradient under clipping**. In Section 2, we quantify off-policy degree via the variance of the IS ratio and derive a closed-form characterization of the conditional gradient expectation. A key consequence is a **gradient dominance reversal**: in near on-policy regimes, low-probability tokens can dominate the effective update, while in sufficiently off-policy regimes their IS ratios become highly dispersed and are frequently clipped, causing higher-probability tokens to dominate. This reversal unifies the conflicting empirical findings in prior work.

Beyond resolving this contradiction, we ask more generally: what governs (i) which tokens contribute to the update, and (ii) how the update direction deviates from the on-policy ideal? In Section 3, we provide a component-wise analysis of RLVR updates along two dimensions: the *magnitude* of the effective update and its *directional bias* relative to the on-policy gradient. We show that token probability, advantage sign/magnitude, and IS-ratio clipping interact to shape both. These results highlight a fundamental mismatch in standard practice: a single global clipping window is not aligned with the highly heterogeneous IS-ratio dispersion across token groups and training regimes, and can also substantially rotate the optimization direction.

Motivated by these insights, we propose **Adaptive Clip Policy Optimization (ACPO)** in Section 4.2. ACPO replaces the fixed global clipping range with group-specific (bin-wise) clipping thresholds that adapt to heterogeneous IS-ratio dispersion across token groups, reducing uneven clipping and mitigating clipping-induced gradient bias. Empirically, ACPO consistently outperforms strong baselines (e.g., DAPO (Yu et al., 2025) and CISPO (Chen et al., 2025a)) across three reasoning tasks under both near on-

policy and off-policy training regimes. In summary, our main contributions are as follows:

- Analysis of clipped RLVR: off-policy degree shifts gradient expectation via IS-ratio dispersion, causing gradient dominance reversal and reconciling prior heuristics.
- Joint effects of token probability, advantage, and IS-ratio clipping on update magnitude/direction, explaining why a single global clip can be mismatched.
- ACPO: variance-aware adaptive clipping with consistent gains over strong baselines across model scales, tasks, and both near on-policy and off-policy regimes.

## 2. Theoretical Analysis of Empirical Contradictions

### 2.1. Preliminary

In contrast to traditional actor-critic algorithms such as Proximal Policy Optimization (PPO) (Schulman et al., 2017), Group Relative Policy Optimization (GRPO) (Shao et al., 2024) removes the critic (of comparable size to the policy) and estimates advantages by standardizing rewards within a sampled group. For each prompt $q \sim P(Q)$, the policy $\pi_\theta$ samples $G$ responses $\{o_i\}_{i=1}^G$, each scored by a rule-based reward $r_i = R(o_i)$. The relative advantage is $A_i = \frac{r_i - \text{mean}(\{r_j\}_{j=1}^G)}{\text{std}(\{r_j\}_{j=1}^G)}$, where $\text{mean}$ and $\text{std}$ denote the sample mean and standard deviation, respectively. The objective is:

$$J_{GRPO}(\theta) = \mathbb{E}_{q \sim P(Q), \{o_i\}_{i=1}^G \sim \pi_{\theta_{old}}(O|q)} \frac{1}{G} \sum_{i=1}^G \frac{1}{|o_i|} \sum_{t=1}^{|o_i|} \quad (1)$$

$$\{\min[\rho_{i,t}A_i, \text{clip}(\rho_{i,t}, 1 - \epsilon_l, 1 + \epsilon_h)A_i]\}$$

where $\rho_{i,t} = \frac{\pi_\theta(o_{i,t}|q, o_{i,<t})}{\pi_{\theta_{old}}(o_{i,t}|q, o_{i,<t})}$ is the importance sampling

(IS) ratio, used to correct for off-policy updates. The clipping function $clip(\cdot, 1 - \epsilon_l, 1 + \epsilon_h)$ restricts this ratio to $[1 - \epsilon_l, 1 + \epsilon_h]$ for stable updates. GRPO typically uses symmetric clipping ($\epsilon_l = \epsilon_h$), while DAPO (Yu et al., 2025) adopts asymmetric clipping ($\epsilon_h > \epsilon_l$) to mitigate entropy collapse. Following recent practice (Yu et al., 2025; Su et al., 2025; Chen et al., 2025a), we omit the KL penalty.

## 2.2. Analysis of Divergent Empirical Findings

### 2.2.1. CONTRADICTORY EMPIRICAL RESULTS

The limitations of a heuristic-driven approach are highlighted by recent, conflicting findings. Wang et al. (2025b) report faster convergence and superior accuracy by selectively updating high-entropy tokens, as these tokens are pivotal for reasoning. In contrast, Yang et al. (2025b) argue against amplifying the influence of low-probability tokens, showing that exclusively updating them leads to slow convergence and suboptimal performance. Their work suggests that the large, high-variance gradients from these tokens can destabilize training. This creates a paradox, as high-entropy tokens are often those with low generation probabilities (Guo et al., 2025b; Zheng et al., 2025b). That two strategies targeting overlapping token populations yield divergent outcomes suggests that a critical factor is being overlooked. This puzzling discrepancy motivates our investigation into the fundamental structure of the policy gradient.

### 2.2.2. FINDING: DEGREE OF OFF-POLICY IS THE KEY

The key experimental difference between Yang et al. (2025b); Wang et al. (2025b) is the number of updates per rollout: 2 vs. 16. More updates push the current policy further away from the behavior policy that generated the rollouts, increasing the degree of off-policy. Because Eq. (1) uses a clipped IS ratio, off-policy degree affects not only the magnitude of token gradients but also which tokens survive clipping and contribute any update. Therefore, we analyze the **expected effective gradient under clipping**.

To quantify off-policy degree at the mechanism level, we track how dispersed the IS ratio is. Specifically, we model the conditional IS-ratio variance given the behavior-policy probability, $\sigma_\rho^2(\pi) = \text{Var}[\rho_{i,t} \mid \pi_{i,t} = \pi]$, and later aggregate it into a scalar proxy for the overall off-policy degree.

### 2.2.3. DETAILED ANALYSIS

**Overview.** We characterize which tokens dominate the effective GRPO update after clipping by studying a token-level update coefficient and its conditional expectation given the token's behavior-policy probability. Let $\pi_{i,t} \triangleq \pi_{\theta_{\text{old}}}(o_{i,t} \mid q, o_{i,<t})$ for token position $t$ in sampled sequence $i$. We show that increasing off-policy-ness (e.g., more updates per rollout) can reverse which probability region dom-

inates the effective gradient, yielding a **Gradient Dominance Reversal** that reconciles Yang et al. (2025b); Wang et al. (2025b).

**Gradient of GRPO.** To connect our analysis to the GRPO objective, we start from its gradient and express it as a sum of token-level contributions:

$$\nabla_\theta J_{\text{GRPO}}(\theta) = \mathbb{E}_{q \sim P(Q), \{o_i\}_{i=1}^G \sim \pi_{\theta_{\text{old}}}(O|q)} \frac{1}{G} \sum_{i=1}^{G} \frac{1}{|o_i|} \sum_{t=1}^{|o_i|} g_{i,t}(\theta),$$

$$g_{i,t}(\theta) \triangleq \rho_{i,t} A_i \cdot \mathbb{I}_{\text{clip}}(\rho_{i,t}, A_i) \cdot \nabla_\theta \log \pi_\theta(o_{i,t} \mid q, o_{i,<t}),$$

$$(2)$$

The clipping indicator is

$$\mathbb{I}_{\text{clip}}(\rho_{i,t}, A_i) = \begin{cases} 0 & A_i > 0, \ \rho_{i,t} > 1 + \epsilon_h, \\ 0 & A_i < 0, \ \rho_{i,t} < 1 - \epsilon_l, \\ 1 & \text{otherwise.} \end{cases} \quad (3)$$

**From parameter gradients to an analyzable logit coefficient.** Directly analyzing $g_{i,t}(\theta)$ at the parameter level is difficult because it entangles model Jacobians. Instead, we analyze the initial update signal at the pre-softmax logits $z_{i,t}$ and denote the sampled vocab index by $k = o_{i,t}$. For softmax, $\frac{\partial \log \pi_\theta(k|q, o_{i,<t})}{\partial z_{i,t,k}} = 1 - \pi_\theta(k \mid q, o_{i,<t})$. Under the standard small-step regime, $\pi_\theta(k \mid q, o_{i,<t}) \approx \pi_{\theta_{\text{old}}}(k \mid q, o_{i,<t})$, so the token-wise effective logit-gradient coefficient can be approximated as

$$G_{i,t} \triangleq (1 - \pi_{i,t}) \cdot \rho_{i,t} \cdot A_i \cdot \mathbb{I}_{\text{clip}}(\rho_{i,t}, A_i). \quad (4)$$

Here we use $\pi \in (0, 1)$ to denote a specific value of random variable $\pi_{i,t}$ when conditioning. We study the conditional expectation $\mathbb{E}[G_{i,t} \mid \pi_{i,t} = \pi]$, abbreviated as $\mathbb{E}[G \mid \pi]$. The expectation is over the sampled prompts (and the induced randomness in $A_i$ and $\rho_{i,t}$), conditioned on $\pi_{i,t} = \pi$.

In Eq. 4, a token contributes only if $\rho_{i,t}$ lies in the clipping interval $[1 - \epsilon_l, \ 1 + \epsilon_h]$ (with the boundary chosen based on the sign of $A_i$). Thus, analyzing $\mathbb{E}[G_{i,t} \mid \pi_{i,t} = \pi]$ reduces to characterizing how often $\rho_{i,t}$ falls inside this interval under the conditioning.

Conditioned on $\pi_{i,t} = \pi$, $\rho_{i,t}$ remains random because the current-policy probability $\pi_\theta(o_{i,t} \mid q, o_{i,<t})$ varies across prompts and evolves across updates. We denote this variability by the conditional distribution $p(\rho \mid \pi)$. We denote this variability by the conditional distribution $p(\rho \mid \pi)$ and summarize its spread by the conditional variance

$$\sigma_\rho^2(\pi) \triangleq \text{Var}[\rho_{i,t} \mid \pi_{i,t} = \pi]. \quad (5)$$

We will treat $\sigma_\rho^2(\pi)$ as our local, token-group-specific measure of off-policy degree. As training becomes more off-policy, $p(\rho \mid \pi)$ typically becomes more dispersed and

$\sigma_\rho^2(\pi)$ increases, which changes the expected amount of clipping and thus reshapes $\mathbb{E}[G \mid \pi]$.

To make this dependence analyzable, we approximate $\rho_{i,t} \mid (\pi_{i,t} = \pi)$ by a Gaussian distribution $\mathcal{N}(1, \sigma_\rho^2(\pi))$. The mean is set to 1 because, over random prompts and advantages, a small stochastic update can increase or decrease a token's probability. The key is to characterize how $\sigma_\rho^2(\pi)$ depends on $\pi$ under a small update step. Using a first-order approximation (Appendix B.1), we obtain:

**Lemma 2.1** (Variance of Importance Sampling Ratio). *Consider a single-step gradient update from policy $\pi_{old}$ to $\pi$. For a token with probability $\pi$, the IS-ratio variance admits the approximation*

$$\sigma_\rho^2(\pi) = \kappa^2 (1 - \pi)^2 + O(\kappa^3), \qquad (6)$$

*where $\kappa$ summarizes the per-step logit perturbation scale. The higher-order term $O(\kappa^3)$ is negligible under the standard small-step assumption.*

Eq. (6) shows that $\sigma_\rho^2(\pi)$ grows rapidly as $\pi$ decreases, implying a much wider IS-ratio distribution for low-probability tokens and hence a higher likelihood of being clipped. Figure 2(a) provides an empirical visualization of this principle. The distribution of the IS ratio for low-probability tokens is shown to have a much larger spread and numerous outliers compared to that of high-probability tokens. We further fit $\mathrm{Var}(\rho)$ against $(1 - \pi)$ and obtain an exponent of $1.90 \pm 0.14$, closely matching the theoretical value of 2 (Appendix Figure 6).

**Expectation of Gradient.** Lemma 2.1 characterizes how the IS-ratio variance changes with the behavior probability $\pi$. We next translate this into an analytical approximation for the conditional expectation $\mathbb{E}[G \mid \pi]$, i.e., the expected *effective* logit-gradient coefficient after clipping. To obtain a closed form, we assume the advantage is Gaussian, $A \sim \mathcal{N}(0, \sigma_A^2)$, and use the Gaussian approximation for $\rho \mid \pi$. The full derivation is provided in Appendix B.2.

**Proposition 2.2** (Conditional Gradient Expectation). *Fix a sampled token whose behavior-policy probability is $\pi \in (0, 1)$. Its conditional expectation $\mathbb{E}[G \mid \pi]$ can be expressed as*

$$\mathbb{E}[G \mid \pi] = (1 - \pi) \frac{\sigma_A}{\sqrt{2\pi}} \cdot F_{\epsilon_h, \epsilon_l}(\sigma_\rho(\pi)), \qquad (7)$$

*where $\sigma_\rho(\pi) = \sqrt{\sigma_\rho^2(\pi)}$ and*

$$
\begin{aligned}
F_{\epsilon_h, \epsilon_l}(\sigma_\rho(\pi)) &= \Phi\left(\frac{\epsilon_h}{\sigma_\rho(\pi)}\right) - \Phi\left(\frac{\epsilon_l}{\sigma_\rho(\pi)}\right) \\
&\quad - \sigma_\rho(\pi)\left[\phi\left(\frac{\epsilon_h}{\sigma_\rho(\pi)}\right) + \phi\left(\frac{\epsilon_l}{\sigma_\rho(\pi)}\right)\right],
\end{aligned} \qquad (8)
$$

*and $\Phi(\cdot)$ and $\phi(\cdot)$ are the CDF and PDF of the standard normal distribution, respectively.*

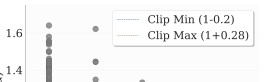
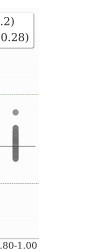
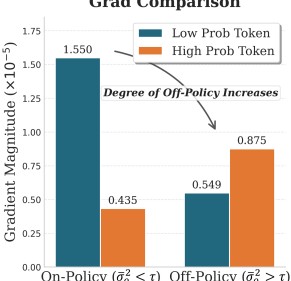

*Figure 2.* Empirical validation of importance sampling ratio variance and gradient dominance. (a) Importance sampling ratios show higher variance for low-probability tokens. (b) As off-policy degree increases, gradient dominance shifts from low- to high-probability tokens.

The factor $(1 - \pi)$ recovers the familiar on-policy scaling of the logit gradient, while $F_{\epsilon_h, \epsilon_l}(\sigma_\rho(\pi))$ captures the suppression induced by clipping. In particular, $F_{\epsilon_h, \epsilon_l}$ is governed by the standardized clipping thresholds $\frac{\epsilon_h}{\sigma_\rho(\pi)}$ and $\frac{\epsilon_l}{\sigma_\rho(\pi)}$.

**Two Different Gradient Dominances.** Proposition 2.2 enables us to compare the expected update contributions from different token populations under clipping. We partition the token-probability space into a low-probability interval $I_L = [0, p_L]$ and a high-probability interval $I_H = [p_H, 1)$, for some thresholds $0 < p_L < p_H < 1$. We define the group-level expected gradient signal as the conditional average of $\mathbb{E}[G \mid \pi]$ over the rollout distribution $p(\pi)$ within each interval:

$$\bar{G}_L = \frac{1}{\mathbb{P}(\pi \in I_L)} \int_0^{p_L} \mathbb{E}[G \mid \pi]\, p(\pi)\, d\pi, \qquad (9)$$

$$\bar{G}_H = \frac{1}{\mathbb{P}(\pi \in I_H)} \int_{p_H}^1 \mathbb{E}[G \mid \pi]\, p(\pi)\, d\pi, \qquad (10)$$

where $\mathbb{P}(\pi \in I) = \int_I p(\pi)\, d\pi$. We compare the magnitudes $|\bar{G}_L|$ and $|\bar{G}_H|$, i.e., we first take expectations within each group and then take absolute values, to identify which population dominates the expected update signal (Detailed derivations are in Appendix B.3).

With these group-level quantities in place, we can now characterize how the dominant contribution to the expected update signal changes with the degree of off-policy induced by repeated updates.

**Corollary 2.3** (Gradient Dominance Reversal). *For the degree of off-policy $\bar{\sigma}_\rho^2$, there exists a threshold $\tau > 0$ that dictates a reversal in which token population dominates the gradient expectation:*

1. *__Near On-Policy Regime__ ($\bar{\sigma}_\rho^2 < \tau$): Clipping is rare. The gradient magnitude is dominated by $(1 - \pi)$, yielding $|\bar{G}_L| > |\bar{G}_H|$; low-probability tokens drive updates.*
2. *__Sufficiently Off-Policy Regime__ ($\bar{\sigma}_\rho^2 > \tau$): Low-probability tokens have higher IS-ratio variance as $\pi$*

*decreases (scaling with $(1-\pi)^2$; Lemma 2.1), so they are frequently clipped and $|\bar{G}_H| > |\bar{G}_L|$; high-probability tokens dominate.*

We validate this reversal in Fig. 2(b), tracking gradient norms of Qwen2.5-7B (Yang et al., 2024) trained on math reasoning with GRPO. Near on-policy (left), low-probability tokens yield much larger gradients than high-probability tokens. As off-policy degree crosses the threshold (right), low-probability contributions collapse under clipping, and high-probability tokens dominate learning.

**Resolving the Empirical Contradiction.**

This dominance reversal unifies the conflicting observations in prior work. Yang et al. (2025b) operate in a near on-policy regime, where low-probability tokens naturally dominate the effective gradient (Figure 2(b)) but also introduce high variance; suppressing them stabilizes learning. Wang et al. (2025b) operate in a more off-policy regime, where low-probability tokens exhibit much larger IS-ratio variance (Figure 2(a)) and are frequently clipped, so the remaining contributors within a high-entropy selection are comparatively stable and informative.

## 3. Key Factors in LLM RL Updates

Section 2 shows that seemingly conflicting token-update heuristics can be reconciled once we analyze the effective update under clipped importance sampling: the off-policy degree changes the IS-ratio dispersion, which in turn changes which tokens survive clipping and dominate the expected gradient. To turn this insight into principled algorithm design, we further ask: beyond token probability, what factors control (i) which tokens contribute to the update, and (ii) how the update direction deviates from the on-policy ideal? This section provides a component-wise analysis of the GRPO update that will directly motivate adaptive clipping in Section 4.2.

### 3.1. Analytical Framework: magnitude vs. direction

We analyze GRPO updates from two perspectives.

**Magnitude (who dominates the update).** Following Section 2, we use the conditional gradient expectation $\mathbb{E}[G \mid \pi]$ (Proposition 2.2) as the primary quantity to characterize *effective* token contributions after clipping. This captures not only per-token gradient scaling, but also the probability of being clipped.

**Direction (how the optimization trajectory changes).** Clipping is not merely a variance-reduction device: by zeroing out subsets of token-level updates, it can change the composition of the gradient and thus rotate the optimization direction even when the overall norm is similar. To quantify such directional changes in high dimensions, we

adopt a layer-wise geometric measure. For a layer gradient matrix $W \in \mathbb{R}^{m \times n}$, we compute its SVD $W = U\Sigma V^\top$ and represent the dominant update directions by the subspace spanned by the top-$k$ right singular vectors $V_k$. Given two gradients with subspace bases $Q^A$ and $Q^B$, we measure their alignment using principal angles:

$$\theta_i = \arccos(s_i), \quad i = 1, \ldots, k, \tag{11}$$

where $s_i$ are the singular values of $(Q^A)^\top Q^B$. We report mean angles over the top-$k$ directions (details in Appendix C).

### 3.2. Key Factors

Eq. 4 shows that GRPO's *effective* gradient is jointly determined by (i) token-wise sensitivity and probability-dependent IS variability, (ii) the advantage magnitude and sign, and (iii) IS-ratio clipping that gates off-policy contributions. We analyze each factor through both gradient expectation $\mathbb{E}[G \mid \pi]$ (magnitude dominance) and subspace alignment (directional distortion).

**Token properties: tail tokens are strong but fragile contributors.** Low-probability tokens have large base sensitivity via $(1 - \pi_k)$, making them potentially dominant contributors in near on-policy regimes (Corollary 2.3). However, Lemma 2.1 shows that their IS-ratio variance grows rapidly as $\pi_k$ decreases, increasing their clipping probability as training becomes more off-policy. Empirically, gradients restricted to low-probability tokens exhibit disproportionately large norms and strong alignment with the full update (Appendix E.1), confirming that the learning signal is concentrated in the distribution tail, but that this signal is also the most susceptible to being filtered out by clipping. This clarifies why entropy-/probability-based heuristics can both help: they are implicitly trading off signal strength (tail tokens) against stability under clipping.

**Advantage sign: negative-advantage samples steer the update direction.** While the advantage magnitude scales the update, its sign changes the qualitative effect of the update on the policy. Positive-advantage updates ($A > 0$) reinforce sampled actions, whereas negative-advantage updates ($A < 0$) penalize specific actions and effectively redistribute probability mass across many alternatives. As a result, the gradient component from $A < 0$ samples can dominate the directional structure even when the norms are comparable. Consistent with this mechanism, we find that the negative-advantage gradient subspace aligns substantially better with the final update direction than the positive-advantage one (Appendix E.2). This highlights that RLVR optimization is often driven more by error correction than by simply reinforcing already-good trajectories, and it implies that any gating mechanism (e.g., clipping) that disproportionately suppresses $A < 0$ updates can significantly rotate the learn-

ing direction.

**Importance sampling and clipping: stability comes with directional distortion.** Importance sampling enables reuse of rollouts under policy drift, but the variance of $\rho_k$ necessitates clipping for stable training. Crucially, clipping gates token updates through $\mathbb{I}_{\text{clip}}(\rho_k, A)$, which changes the effective sample set and can therefore distort the update direction. We empirically compare an on-policy gradient (aggregating samples into a single step) with the standard off-policy clipped gradient and observe a large directional divergence ($\approx 47°$) despite similar magnitudes (Appendix E.3). This indicates that clipping does more than control variance: by down-weighting or discarding tokens with extreme IS ratios, it changes which tokens drive learning and can therefore shift the update direction away from the on-policy gradient, even when the gradient norm is similar.

### 3.3. Summary and design implication

Our analysis yields two algorithm design insights.

**Uniform clipping is mismatched to heterogeneous IS-ratio dispersion.** Since IS-ratio variance depends strongly on token probability (Lemma 2.1) and changes with the off-policy degree, a single global clipping window makes the probability of being clipped highly uneven across token groups, systematically over-clipping some groups (typically low-probability tokens) while letting others dominate the effective update.

**Clipping affects not only variance but also the optimization trajectory.** By changing which token updates survive, clipping can substantially rotate the gradient direction relative to the on-policy ideal, even when the overall gradient norm is preserved. This suggests that improving RLVR updates is not only about stabilizing magnitudes, but also about controlling which tokens steer the direction.

**Implication: clipping should be adaptive to the local policy shift.** Together, these results suggest that the clipping window should be calibrated per token group according to how off-policy that group is, rather than being fixed globally. A natural, directly observable proxy is the within-group IS-ratio variance, $\sigma_\rho^2(\pi)$. This motivates the adaptive, variance-aware clipping strategy introduced in Section 4.2.

## 4. Adaptive Clip Policy Optimization

### 4.1. Motivation

GRPO reuses rollout trajectories via importance-sampling (IS) ratios and stabilizes learning with clipping. The gradient of the clipped objective (Eq. 2, cf. Eq. 4) implies that a token contributes to the update only when its IS ratio lies within the clipping window, so learning depends critically on how dispersed IS ratios are across tokens. Lemma 2.1

provides an expression for the IS-ratio variance, implying substantially larger dispersion for low-probability tokens. With a single global clipping range, this leads to uneven clipping rates across token groups. Proposition 2.2 and Corollary 2.3 further show that such uneven clipping biases the expected gradient and can even change which token groups dominate the update, consistent with the update-direction distortions observed in Section 3. These results motivate bin-specific clipping thresholds $\epsilon_b$, scaled by the within-bin standard deviation of IS ratios estimated from each update batch, to reduce uneven clipping across groups and mitigate clipping-induced gradient bias.

### 4.2. Adaptive Clip Policy Optimization (ACPO)

ACPO replaces the single global clipping range in Eq. 1 with a bin-specific range determined by the behavior-token probability. Given an update mini-batch (pooling all tokens in the update batch), we assign each token $(i, t)$ to one of $B$ equal-width bins on $[0, 1]$ via

$$b(i, t) = \min\{B, \lfloor B \cdot \pi_{\theta_{\text{old}}}(o_{i,t} \mid q, o_{i,<t}) \rfloor + 1\}. \quad (12)$$

For each bin $b$, we compute the within-bin dispersion of IS ratios on this mini-batch and set the clipping range as

$$\sigma_b = \text{Std}(\{\rho_{i,t} : \mathcal{B}(i, t) = b\}),$$
$$\epsilon_b = \epsilon_{\text{base}} + \alpha\, \sigma_b. \quad (13)$$

Applying token-wise clipping by using $\epsilon_{\mathcal{B}(i,t)}$ in Eq. 1. The objective is

$$J_{\text{ACPO}}(\theta) = \mathbb{E}_{q \sim P(Q),\, \{o_i\}_{i=1}^G \sim \pi_{\theta_{\text{old}}}(O|q)} \frac{1}{G} \sum_{i=1}^{G} \frac{1}{|o_i|} \sum_{t=1}^{|o_i|}$$
$$\min\left[\rho_{i,t} A_i,\; \text{clip}(\rho_{i,t}, 1 - \epsilon_{\mathcal{B}(i,t)}, 1 + \epsilon_{\mathcal{B}(i,t)}) A_i\right], \quad (14)$$

See Algorithm 1 in Appendix F.

## 5. Experiment

### 5.1. Experimental Setup

**Tasks and Models.** We evaluate ACPO on three reasoning tasks spanning different domains: ORZ-57K (Hu et al., 2025) for mathematical problem solving, HiTab (Cheng et al., 2022) for tabular question answering, and Countdown (Jackson, 2025) for arithmetic-based puzzles. Experiments are conducted with Qwen2.5-3B, Qwen2.5-3B-Instruct, and Qwen2.5-7B (Yang et al., 2024) to examine scalability across model sizes.

**Baselines.** We compare against baselines representing different design choices: (1) **DAPO** (Yu et al., 2025): asymmetric clipping with $\epsilon_h > \epsilon_l$; (2) **CISPO** (Chen et al., 2025a): clips only IS ratios rather than gating token updates;

*Table 1.* Countdown/HiTab results. Best validation reward (N-OnP./OffP.), numbers are mean accuracy over 4 evaluations per benchmark.

*(a)* Performance of model trained on the Countdown dataset.

| Method | Qwen2.5-3B-Instruct | |
|---|---|---|
| | **N-OnP.** | **OffP.** |
| Base Model | 6.90 | |
| AR-Lopti | 65.82 | 57.41 |
| High-Entropy | 70.58 | 73.76 |
| Low-Entropy | 61.81 | 74.04 |
| DAPO | 73.27 | 74.38 |
| CISPO | 74.25 | 55.12 |
| ACPO | **75.74** | **76.27** |

*(b)* Performance of model trained on the HiTab dataset.

| Method | Qwen2.5-3B | | Qwen2.5-7B | |
|---|---|---|---|---|
| | **N-OnP.** | **OffP.** | **N-OnP.** | **OffP.** |
| Base Model | 17.50 | | 31.25 | |
| AR-Lopti | 45.50 | 43.33 | 68.92 | 66.58 |
| High-Entropy | 45.00 | 43.50 | 64.33 | 61.83 |
| Low-Entropy | 43.17 | 43.00 | 67.00 | 65.08 |
| DAPO | 44.33 | 44.00 | 68.42 | 65.75 |
| CISPO | 43.92 | 44.83 | 69.17 | 65.67 |
| ACPO | **45.50** | **46.42** | **69.83** | **66.58** |

*Table 2.* Results of Qwen2.5-7B model trained on the ORZ-57K dataset. For AIME24/25 we report mean@32; others are mean@8.

| Method | Minerva | | Math500 | | AMC | | AIME24 | | AIME25 | | Olympiad | | Avg. | |
|---|---|---|---|---|---|---|---|---|---|---|---|---|---|---|
| | **N-OnP.** | **OffP.** | **N-OnP.** | **OffP.** | **N-OnP.** | **OffP.** | **N-OnP.** | **OffP.** | **N-OnP.** | **OffP.** | **N-OnP.** | **OffP.** | **N-OnP.** | **OffP.** |
| Base Model | 13.32 | | 41.48 | | 26.20 | | 0.83 | | 0.62 | | 18.02 | | 16.75 | |
| AR-Lopti | 29.14 | 25.14 | 73.75 | 63.92 | 44.88 | 34.94 | 15.21 | 8.75 | 3.96 | 2.08 | 33.60 | 24.00 | 33.42 | 26.47 |
| High-Entropy | 29.08 | 29.27 | 76.88 | 75.22 | 50.75 | 46.39 | 14.37 | 14.06 | 7.81 | **10.62** | 33.54 | 34.54 | 35.41 | 35.02 |
| Low-Entropy | 28.72 | 25.05 | 75.85 | 72.47 | 48.95 | 40.81 | 16.04 | 11.15 | 9.48 | 4.58 | 34.69 | 26.61 | 35.62 | 30.11 |
| DAPO | 29.92 | 30.47 | 78.05 | 76.72 | 51.20 | 46.84 | **19.69** | **18.02** | 11.88 | 6.56 | 41.44 | 37.20 | 38.70 | 35.97 |
| CISPO | 30.51 | 29.55 | 78.57 | 76.30 | 49.25 | 46.08 | 18.65 | 13.44 | 10.00 | 8.33 | 40.61 | 39.10 | 37.93 | 35.47 |
| ACPO (Ours) | **31.02** | **31.66** | **79.78** | **80.48** | **54.37** | **51.20** | 16.77 | 16.35 | **14.37** | 10.10 | **42.83** | **42.17** | **39.86** | **38.66** |

(3) **High/Low-Entropy** (Wang et al., 2025b): uses only high-entropy (top 20%) or low-entropy (bottom 80%) tokens for gradient updates; (4) **AR-Lopti** (Yang et al., 2025b): reduces low-probability token dominance via scaled token-wise contributions.

**Training Regimes.** Following Yang et al. (2025b); Wang et al. (2025b), we evaluate under **near on-policy** (N-OnP., 2 updates/rollout) and **off-policy** (OffP., 16 updates/rollout) to test robustness across policy divergence levels.

**Evaluation.** For ORZ models, we evaluate on six math benchmarks (Minerva (Dyer & Gur-Ari, 2022), Math500 (Hendrycks et al., 2021), AMC2023, AIME24/25, OlympiadBench (He et al., 2024)), reporting avg@32 for AIME and avg@8 for others. HiTab and Countdown use avg@4 test accuracy. Statistical significance uses one-tailed Welch's $t$-test ($\alpha = 0.05$); full $p$-values in Appendix K.

**Hyperparameters.** ACPO uses $B = 5$ probability bins, $\alpha = 3$, and $\epsilon_{\text{base}} = 0.2$. All methods share the same base configuration (learning rate $10^{-6}$, batch size 512, temperature 1.0). Hyperparameter sensitivity analysis is provided in Appendix I.1, and full training configurations for all methods are detailed in Appendix H.

### 5.2. Main Results

**Consistent improvements across settings.** Tables 1–2 present results across three reasoning tasks (ORZ, HiTab,

Countdown), multiple model scales (3B, 7B), and two training regimes (near on-policy and off-policy). ACPO achieves the best average rank of 1.33 across all 18 task-regime configurations, with 12/18 first-place and 5/18 second-place finishes, compared to DAPO (avg rank 2.67, 3/18 wins) and CISPO (avg rank 3.11, 0/18 wins). One-tailed Welch's $t$-tests show that ACPO's gains are statistically significant ($p < 0.05$) on most benchmarks (Appendix K).

**Cross-regime stability.** ACPO remains stable when moving from near on-policy to off-policy, with only a 0.8% performance drop (Figure 3). Among baselines, CISPO and AR-Lopti suffer substantial degradation ($-6.0\%$ and $-5.0\%$), while DAPO shows moderate drop ($-1.2\%$). High/Low-Entropy appear stable but with significantly lower overall performance. CISPO's instability stems from clipping only IS-ratio values without gating gradient updates—tokens with extreme IS ratios still contribute misleading gradients. AR-Lopti's degradation suggests that suppressing low-probability tokens becomes counterproductive when gradient dominance reverses under off-policy training (Corollary 2.3). ACPO avoids these failure modes by adaptively calibrating thresholds to local IS-ratio dispersion.

**Takeaway.** These results validate our analysis: by handling heterogeneous IS-ratio variance across token-probability bins, ACPO achieves consistent improvements and cross-regime stability, supporting the claim that effective RLVR requires adapting *which tokens survive clipping* as off-policy

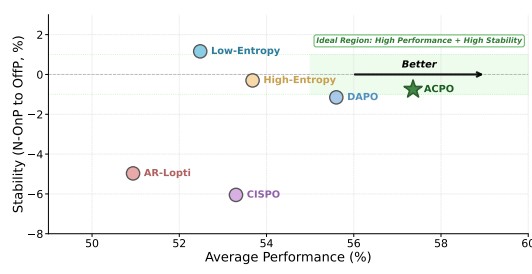

*Figure 3.* Performance vs. cross-regime stability.

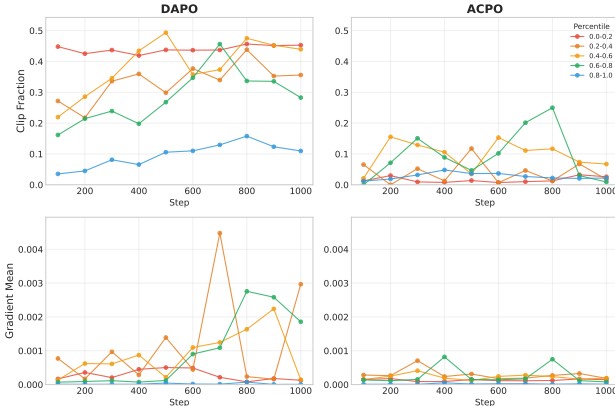

*Figure 4.* Training dynamics under off-policy training across five probability bins. **Top:** Clip fraction over training steps. DAPO's fixed threshold yields heterogeneous rates; ACPO achieves uniform, low rates. **Bottom:** Gradient std over training steps. DAPO exhibits unstable spikes; ACPO remains stable throughout.

degree changes rather than relying on fixed heuristics.

### 5.3. Analysis

To understand the mechanisms underlying ACPO's stability, we analyze token-level clipping behavior and gradient dynamics during off-policy training on ORZ. Figure 4 compares DAPO and ACPO across five probability bins.

**Clip fraction.** DAPO exhibits heterogeneous clipping rates across bins: low-probability tokens (0.0–0.4) are clipped at 35–50%, while high-probability tokens (0.8–1.0) at only 5–15%. This disparity arises because fixed thresholds cannot accommodate the varying IS-ratio variance—tokens with inherently higher variance exceed the threshold more frequently, leading to over-clipping that discards potentially informative gradient signals. ACPO achieves more uniform clip rates across all bins (<20%) by setting per-bin thresholds as $\epsilon_b = \epsilon_{\text{base}} + \alpha \cdot \sigma_b$, which naturally accommodates the heterogeneous variance structure.

**Gradient variance.** DAPO shows highly unstable gradient variance, with standard deviation spiking to over 0.02 for mid-probability bins (0.2–0.4) around step 600–800. ACPO maintains stable gradient variance below 0.005 throughout training for all bins, confirming that adaptive clipping provide consistent regularization across different token groups and training stages.

These dynamics explain ACPO's improved performance and stability. Appendices I.2–J confirm that gains stem from adaptive calibration rather than simply using different threshold values, with negligible computational overhead.

### 6. Related Works

**Reinforcement Learning for Large Language Models.** The application of reinforcement learning to LLMs has transitioned from preference alignment via RLHF (Ouyang et al., 2022) to directly optimizing reasoning abilities with Reinforcement Learning from Verifiable Rewards (RLVR) (Lambert et al., 2024). This paradigm uses objective feedback signals, such as mathematical correctness, to train models. Pioneered by early work like OpenAI's o1 (Jaech et al., 2024) and advanced by algorithms such as GRPO (Shao et al., 2024), RLVR has become a central methodology for developing state-of-the-art reasoning agents, demonstrating that complex skills can emerge from outcome-based optimization.

**Token-Level Update Strategies.** The success of RLVR is highly dependent on the token-level update mechanism. A diverse range of heuristic strategies has been proposed, focusing on areas like entropy-based modulation (Wang et al., 2025b; Cui et al., 2025), gradient clipping (Chen et al., 2025a; Zheng et al., 2025a), and advantage function design. However, these methods often yield conflicting conclusions. For instance, a key debate exists on whether to prioritize high-entropy tokens to encourage exploration in reasoning (Wang et al., 2025b) or to suppress them to mitigate high-variance gradients (Yang et al., 2025b). Such contradictions highlight a lack of a unified understanding of RLVR's update dynamics, motivating the need for the systematic, component-wise analysis presented in our work.

### 7. Conclusion

We present a token-level analysis of clipped RLVR updates and identify off-policy degree as the key factor that reconciles prior conflicting heuristics. As off-policy increases, IS-ratio dispersion and clipping reshuffle which tokens dominate the expected effective gradient, revealing the mismatch of uniform clipping. Motivated by this, we propose ACPO with group-specific clipping thresholds. Experiments on 3B/7B models across diverse reasoning tasks show consistent gains over strong baselines.

Experiments across model scales, diverse reasoning tasks, and training regimes demonstrate that ACPO achieves consistent improvements and cross-regime stability, supporting our central thesis that understanding token-level gradient dynamics, not just choosing which tokens to update, is key to effective RLVR.

## Impact Statement

This paper studies the learning dynamics of clipped policy-gradient updates in Reinforcement Learning from Verifiable Rewards (RLVR) for improving LLM reasoning, and proposes Adaptive Clip Policy Optimization (ACPO) to make such updates more stable and effective. We expect the primary positive impact to be improved reliability and predictability of RLVR training, which can support safer deployment of models on verifiable tasks such as mathematical and logical reasoning. By providing a unified explanation for previously contradictory token-level heuristics, our analysis may reduce trial-and-error in algorithm design and encourage more transparent, diagnostic-driven development.

At the same time, capability improvements in reasoning models can be dual-use, potentially lowering the cost of generating persuasive or strategically optimized content. Our contribution is methodological and does not introduce new data sources; experiments rely on public benchmarks and do not require access to personal or sensitive information. We encourage practitioners to pair improved RLVR training with standard safety practices (e.g., controlled deployment settings, red-teaming, and monitoring) and to report unintended behaviors observed during fine-tuning.

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

## A. Use of Large Language Models (LLMs) Statement

We used Large Language Models (LLMs) only as auxiliary tools during the preparation of this paper. Specifically, LLMs were employed for proofreading, grammar correction, and improving the readability of sentences. They were also occasionally used to assist with LaTeX formatting (e.g., tables and figures). All technical ideas, experimental design, analysis, and scientific claims are solely the work of the authors.

The authors take full responsibility for the content of the manuscript, including any text generated or polished by the LLM. We have ensured that the LLM-generated text adheres to ethical guidelines and does not contribute to plagiarism or scientific misconduct.

## B. Detailed Theoretical Derivations

### B.1. Derivation of Lemma 2.1

This appendix derives the relationship between the variance of the importance sampling ratio, $\sigma_\rho^2$, and the token probability $\pi_k$ in Lemma 2.1.

#### B.1.1. A SMALL-STEP APPROXIMATION IN LOGIT SPACE

Let $\pi_k$ be produced by a softmax with temperature $T$ over logits $y$: $\pi_k = \frac{\exp(y_k/T)}{\sum_j \exp(y_j/T)}$. A small update perturbs logits by $\Delta y$.

We focus on the leading (first-order) effect of changing $y_k$ on $\pi_k$:

$$\frac{\partial \pi_k}{\partial y_k} = \frac{\pi_k(1 - \pi_k)}{T}. \tag{15}$$

Thus, for a small step,

$$\Delta \pi_k \approx \frac{\pi_k(1 - \pi_k)}{T} \Delta y_k. \tag{16}$$

The IS ratio for token $k$ is $\rho_k = \frac{\pi_k'}{\pi_k} = 1 + \frac{\Delta \pi_k}{\pi_k}$, so

$$\rho_k \approx 1 + \frac{(1 - \pi_k)}{T} \Delta y_k. \tag{17}$$

#### B.1.2. VARIANCE OF THE IMPORTANCE SAMPLING RATIO

We now estimate the variance of $\rho_k$, denoted as $\sigma_\rho^2$. Starting from the linearized IS ratio in Eq. (17),

$$\rho_k \approx 1 + \frac{(1 - \pi_k)}{T} \Delta y_k, \tag{18}$$

we model the variability of $\rho_k$ as originating from noise in the logit-space update $\Delta y_k$.

**Key assumption (logit-space noise is probability-independent).** We assume the logit perturbation has variance independent of token probability,

$$\mathrm{Var}(\Delta y_k) = \sigma_y^2, \quad \text{independent of } \pi_k. \tag{19}$$

This captures sources such as optimizer stochasticity, minibatch sampling noise, and numerical effects, rather than attributing all variance to the randomness of the advantage.

Under this assumption, taking variance on both sides of the linearization yields

$$\sigma_\rho^2(\pi_k) = \mathrm{Var}(\rho_k) \approx \mathrm{Var}\left(\frac{(1 - \pi_k)}{T} \Delta y_k\right) \tag{20}$$

$$= \frac{(1 - \pi_k)^2}{T^2} \mathrm{Var}(\Delta y_k) \tag{21}$$

$$= \frac{(1 - \pi_k)^2}{T^2} \sigma_y^2. \tag{22}$$

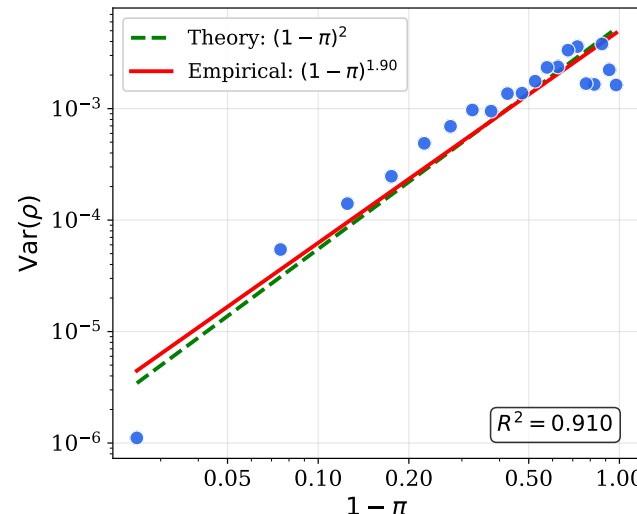

*Figure 5.* Empirical validation of Lemma 2.1 via log-log regression. Green: theoretical scaling $\mathrm{Var}(\rho) \propto (1-\pi)^2$. Red: fitted curve $\mathrm{Var}(\rho) = 5.01 \times 10^{-3}(1-\pi)^{1.90}$ (slope $1.90 \pm 0.14$, $R^2 = 0.91$).

Defining $\kappa := \sigma_y/T$, we arrive at

$$\sigma_\rho^2(\pi_k) \approx \kappa^2(1-\pi_k)^2. \tag{23}$$

This concludes the derivation of Lemma 2.1 under the logit-noise assumption, and shows that $\sigma_\rho^2$ decreases monotonically with $\pi_k$ on $(0,1)$.

### B.1.3. EMPIRICAL VALIDATION OF THE VARIANCE SCALING

To validate the prediction $\mathrm{Var}(\rho) \propto (1-\pi)^2$, we perform a log-log regression on binned tokens. As shown in Figure 6, the fitted exponent is $1.90 \pm 0.14$ with $R^2 = 0.91$ (20 bins; 54,173 tokens), which closely matches the theoretical exponent of 2. The green curve indicates the theoretical scaling, while the red curve shows the empirical fit:

$$\mathrm{Var}(\rho) = 5.01 \times 10^{-3} \cdot (1-\pi)^{1.90}. \tag{24}$$

### B.1.4. EMPIRICAL VALIDATION OF THE VARIANCE SCALING

To validate the prediction $\mathrm{Var}(\rho) \propto (1-\pi)^2$, we perform a log-log regression on binned tokens. As shown in Figure 6, the fitted exponent is $1.90 \pm 0.14$ with $R^2 = 0.91$ (20 bins; 54,173 tokens), which closely matches the theoretical exponent of 2. The green curve indicates the theoretical scaling, while the red curve shows the empirical fit:

$$\mathrm{Var}(\rho) = 5.01 \times 10^{-3} \cdot (1-\pi)^{1.90}. \tag{25}$$

## B.2. Derivation of Proposition 2.2

This section provides a detailed derivation for the expected gradient magnitude, $\mathbb{E}[G|\pi]$, as stated in Proposition 2.2.

### B.2.1. PROBLEM FORMULATION

The policy gradient in a PPO-style objective, ignoring other terms like the KL divergence penalty, is given by:

$$\nabla_\theta J(\theta) = \mathbb{E}_{\pi_{old}}[\nabla_\theta \log \pi_\theta(o_t) \cdot w_t] \tag{26}$$

where $w_t$ is the weighting term after clipping. The policy update for a specific token $k$ with probability $\pi_k$ is proportional to the gradient of the log-probability, $\nabla_\theta \log \pi_\theta(k)$, weighted by $\rho_k \hat{A}_k$, where $\rho_k = \pi_\theta(k)/\pi_{old}(k)$ is the importance sampling

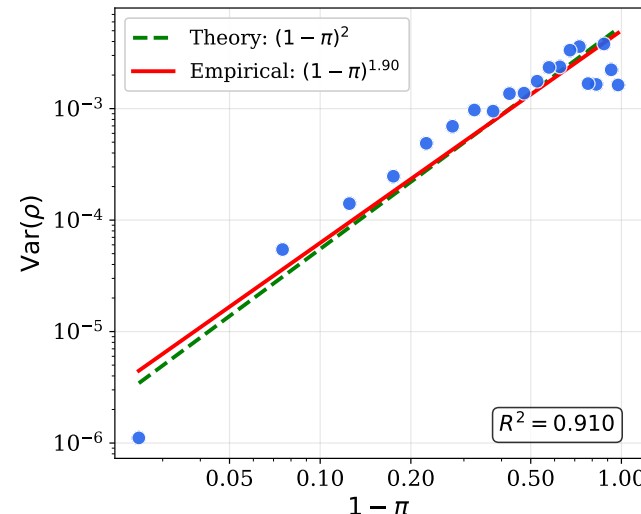

*Figure 6.* Empirical validation of Lemma 2.1 via log-log regression. Green: theoretical scaling $\text{Var}(\rho) \propto (1 - \pi)^2$. Red: fitted curve $\text{Var}(\rho) = 5.01 \times 10^{-3}(1 - \pi)^{1.90}$ (slope $1.90 \pm 0.14$, $R^2 = 0.91$).

ratio and $\hat{A}_k$ is the advantage. The clipping mechanism effectively sets the gradient to zero if the update moves outside the trust region. This can be expressed using an indicator function $\mathbb{I}_{\text{trust}}$:

$$\mathbb{I}_{\text{trust}}(\rho_k, \hat{A}_k) = \begin{cases} 1 & \text{if } (\hat{A}_k > 0 \text{ and } \rho_k \leq 1 + \epsilon_h) \text{ or } (\hat{A}_k < 0 \text{ and } \rho_k \geq 1 - \epsilon_l) \\ 0 & \text{otherwise} \end{cases} \tag{27}$$

The gradient with respect to the pre-softmax logits for token $k$ is proportional to $(1 - \pi_k)$. We define the gradient magnitude component, $G_k$, for a token $k$ as:

$$G_k := (1 - \pi_k)\rho_k \hat{A}_k \mathbb{I}_{\text{trust}}(\rho_k, \hat{A}_k). \tag{28}$$

To derive its expectation, we make the following simplifying assumptions:

1. The advantage $\hat{A}$ is a random variable following a zero-mean normal distribution: $\hat{A} \sim \mathcal{N}(0, \sigma_A^2)$.

2. For a small update step, the importance sampling ratio $\rho$ is approximately normally distributed around 1: $\rho \sim \mathcal{N}(1, \sigma_\rho^2)$, where $\sigma_\rho^2$ is its variance.

3. The random variables $\hat{A}$ and $\rho$ are treated as independent.

### B.2.2. DECOMPOSING THE EXPECTATION

The expectation of $G_k$ can be split into two parts based on the sign of the advantage:

$$\mathbb{E}[G_k] = \mathbb{E}[(1 - \pi_k)\rho_k \hat{A}_k \mathbb{I}_{\text{trust}}] \tag{29}$$

$$= (1 - \pi_k) \left( \mathbb{E}[\rho_k \hat{A}_k \mathbf{1}_{\hat{A}_k > 0, \rho_k \leq 1 + \epsilon_h}] + \mathbb{E}[\rho_k \hat{A}_k \mathbf{1}_{\hat{A}_k < 0, \rho_k \geq 1 - \epsilon_l}] \right) \tag{30}$$

By the independence of $\rho_k$ and $\hat{A}_k$, we can separate the expectations:

$$\mathbb{E}[G_k] = (1 - \pi_k) \left( \mathbb{E}[\hat{A}_k \mathbf{1}_{\hat{A}_k > 0}]\mathbb{E}[\rho_k \mathbf{1}_{\rho_k \leq 1 + \epsilon_h}] + \mathbb{E}[\hat{A}_k \mathbf{1}_{\hat{A}_k < 0}]\mathbb{E}[\rho_k \mathbf{1}_{\rho_k \geq 1 - \epsilon_l}] \right) \tag{31}$$

### B.2.3. COMPUTING TRUNCATED NORMAL EXPECTATIONS

We use the standard formula for the first moment of a truncated normal distribution. If $Z \sim \mathcal{N}(\mu, \sigma^2)$ and $\alpha = (a - \mu)/\sigma$, then:

$$\mathbb{E}[Z\mathbf{1}_{Z \leq a}] = \mu\Phi(\alpha) - \sigma\phi(\alpha) \tag{32}$$

$$\mathbb{E}[Z\mathbf{1}_{Z \geq a}] = \mu(1 - \Phi(\alpha)) + \sigma\phi(\alpha) \tag{33}$$

where $\Phi(\cdot)$ and $\phi(\cdot)$ are the CDF and PDF of the standard normal distribution, respectively.

**For the Advantage $\hat{A} \sim \mathcal{N}(0, \sigma_A^2)$:** Here, $\mu = 0, \sigma = \sigma_A$.

$$\mathbb{E}[\hat{A}\mathbf{1}_{\hat{A}>0}] = 0 \cdot (1 - \Phi(0)) + \sigma_A \phi(0) = \sigma_A \frac{1}{\sqrt{2\pi}} \tag{34}$$

$$\mathbb{E}[\hat{A}\mathbf{1}_{\hat{A}<0}] = 0 \cdot \Phi(0) - \sigma_A \phi(0) = -\sigma_A \frac{1}{\sqrt{2\pi}} \tag{35}$$

**For the IS Ratio $\rho \sim \mathcal{N}(1, \sigma_\rho^2)$:** Here, $\mu = 1, \sigma = \sigma_\rho$. For the upper clip boundary $a_h = 1 + \epsilon_h$, the standardized value is $\alpha_h = \frac{(1+\epsilon_h)-1}{\sigma_\rho} = \frac{\epsilon_h}{\sigma_\rho}$:

$$\mathbb{E}[\rho\mathbf{1}_{\rho \leq 1+\epsilon_h}] = 1 \cdot \Phi(\alpha_h) - \sigma_\rho \phi(\alpha_h) \tag{36}$$

For the lower clip boundary $a_l = 1 - \epsilon_l$, the standardized value is $\alpha_l = \frac{(1-\epsilon_l)-1}{\sigma_\rho} = -\frac{\epsilon_l}{\sigma_\rho}$.

$$\mathbb{E}[\rho\mathbf{1}_{\rho \geq 1-\epsilon_l}] = 1 \cdot (1 - \Phi(\alpha_l)) + \sigma_\rho \phi(\alpha_l) \tag{37}$$
$$= \Phi(-\alpha_l) + \sigma_\rho \phi(-\alpha_l) \quad (\text{since } 1 - \Phi(x) = \Phi(-x) \text{ and } \phi(x) = \phi(-x)) \tag{38}$$
$$= \Phi\left(\frac{\epsilon_l}{\sigma_\rho}\right) + \sigma_\rho \phi\left(\frac{\epsilon_l}{\sigma_\rho}\right) \tag{39}$$

B.2.4. ASSEMBLING THE FINAL EXPRESSION

Let's substitute these results back into Equation 31. For clarity, let $C_A = \frac{\sigma_A}{\sqrt{2\pi}}$.

$$\mathbb{E}[G_k] = (1 - \pi_k)\left[C_A\left(\Phi(\tfrac{\epsilon_h}{\sigma_\rho}) - \sigma_\rho\phi(\tfrac{\epsilon_h}{\sigma_\rho})\right) - C_A\left(\Phi(\tfrac{\epsilon_l}{\sigma_\rho}) + \sigma_\rho\phi(\tfrac{\epsilon_l}{\sigma_\rho})\right)\right] \tag{40}$$
$$= (1 - \pi_k)\frac{\sigma_A}{\sqrt{2\pi}}\left[\Phi\left(\frac{\epsilon_h}{\sigma_\rho}\right) - \Phi\left(\frac{\epsilon_l}{\sigma_\rho}\right) - \sigma_\rho\left(\phi\left(\frac{\epsilon_h}{\sigma_\rho}\right) + \phi\left(\frac{\epsilon_l}{\sigma_\rho}\right)\right)\right] \tag{41}$$

Finally, we substitute the expression for $\sigma_\rho$ from Lemma 2.1, $\sigma_\rho(\pi) = \kappa(1 - \pi)$. For a generic token with probability $\pi$, the expected gradient magnitude becomes:

$$\mathbb{E}[G \mid \pi] = (1 - \pi)\frac{\sigma_A}{\sqrt{2\pi}}\left[\Phi\left(\frac{\epsilon_h}{\sigma_\rho(\pi)}\right) - \Phi\left(\frac{\epsilon_l}{\sigma_\rho(\pi)}\right) - \sigma_\rho(\pi)\left(\phi\left(\frac{\epsilon_h}{\sigma_\rho(\pi)}\right) + \phi\left(\frac{\epsilon_l}{\sigma_\rho(\pi)}\right)\right)\right] \tag{42}$$

This matches the form in Proposition 2.2, where $F(\pi; \kappa, \epsilon_h, \epsilon_l)$ is the term in the brackets. This completes the derivation.

**B.3. Derivation of Corollary 2.3**

This appendix provides a more explicit derivation of the gradient dominance reversal in Corollary 2.3. Our goal is to connect (i) the paper's off-policy degree $\bar{\sigma}_\rho^2$ and (ii) the probability-dependent IS dispersion from Lemma 2.1 to a reversal in the *effective* token contribution after clipping.

B.3.1. SETUP

Recall the token-level effective logit-gradient coefficient (Eq. 4)

$$G_{i,t} = (1 - \pi_{i,t})\,\rho_{i,t}\,A_i\,\mathbb{I}_{\text{clip}}(\rho_{i,t}, A_i).$$

Conditioned on the behavior probability $\pi_{i,t} = \pi$, Proposition 2.2 gives

$$\mathbb{E}[G \mid \pi] = (1 - \pi)\,C_A\,F_{\epsilon_h, \epsilon_l}(\sigma_\rho(\pi)), \tag{43}$$

where $C_A = \frac{\sigma_A}{\sqrt{2\pi}}$ and $\sigma_\rho^2(\pi) = \text{Var}[\rho \mid \pi]$.

We study the group-level averages on $I_L = [0, p_L]$ and $I_H = [p_H, 1)$ (as in the main text):

$$\bar{G}_L := \frac{1}{p_L} \int_0^{p_L} \mathbb{E}[G \mid \pi] \, d\pi, \tag{44}$$

$$\bar{G}_H := \frac{1}{1 - p_H} \int_{p_H}^1 \mathbb{E}[G \mid \pi] \, d\pi. \tag{45}$$

### B.3.2. A GLOBAL UNIFORM-CONTROL ASSUMPTION

Since the main text summarizes off-policy degree by a single scalar $\bar{\sigma}_\rho^2$, to translate $\bar{\sigma}_\rho^2 \to 0$ into a statement about clipping rates we use a standard uniform-control assumption.

**Assumption B.1** (Uniform control by $\bar{\sigma}_\rho^2$). There exists a constant $C \geq 1$ such that for all $\pi \in (0, 1)$,

$$\sigma_\rho^2(\pi) \leq C \, \bar{\sigma}_\rho^2. \tag{46}$$

This assumption is mild in the sense that $\sigma_\rho^2(\pi)$ is finite for each $\pi$ and we only require that its worst case is controlled (up to a constant) by the scalar summary $\bar{\sigma}_\rho^2$.

### B.3.3. STEP 1: NEAR ON-POLICY IMPLIES CLIPPING IS UNIFORMLY RARE

Under the Gaussian approximation $\rho \mid \pi \sim \mathcal{N}(1, \sigma_\rho^2(\pi))$, the probability that a token is clipped is

$$\mathbb{P}[\text{clip} \mid \pi] = \mathbb{P}[\rho > 1 + \epsilon_h \mid \pi] \, \mathbb{P}[A > 0] + \mathbb{P}[\rho < 1 - \epsilon_l \mid \pi] \, \mathbb{P}[A < 0], \tag{47}$$

and since $\mathbb{P}[A > 0] = \mathbb{P}[A < 0] = \frac{1}{2}$ for a symmetric $A \sim \mathcal{N}(0, \sigma_A^2)$,

$$\mathbb{P}[\text{clip} \mid \pi] \leq \tfrac{1}{2} \mathbb{P}[\rho > 1 + \epsilon_h \mid \pi] + \tfrac{1}{2} \mathbb{P}[\rho < 1 - \epsilon_l \mid \pi]. \tag{48}$$

By a standard Gaussian tail bound, for any $\delta > 0$ there exists $\bar{\sigma}_0^2(\delta)$ such that if $\bar{\sigma}_\rho^2 \leq \bar{\sigma}_0^2(\delta)$ then (using Eq. (46))

$$\sup_{\pi \in (0,1)} \mathbb{P}[\text{clip} \mid \pi] \leq \delta. \tag{49}$$

Hence in the near on-policy regime ($\bar{\sigma}_\rho^2$ sufficiently small), clipping is uniformly rare across token probabilities. In this regime, $F_{\epsilon_h, \epsilon_l}(\sigma_\rho(\pi))$ remains close to its unclipped behavior and the base sensitivity factor $(1 - \pi)$ is the dominant source of variation with $\pi$. Consequently, since $(1 - \pi)$ is larger on $I_L$ than on $I_H$, we obtain $|\bar{G}_L| > |\bar{G}_H|$.

### B.3.4. STEP 2: SUFFICIENTLY OFF-POLICY IMPLIES LOW-PROBABILITY TOKENS ARE CLIPPED FIRST

Lemma 2.1 implies that, for a fixed per-step logit-perturbation scale,

$$\sigma_\rho^2(\pi) \propto (1 - \pi)^2, \tag{50}$$

so the conditional IS dispersion is larger for smaller $\pi$. Therefore, as the global off-policy degree $\bar{\sigma}_\rho^2$ increases, the standardized thresholds $\epsilon_h / \sigma_\rho(\pi)$ and $\epsilon_l / \sigma_\rho(\pi)$ shrink more rapidly over $I_L$ than over $I_H$. Equivalently, for a given target clipping rate level $\delta \in (0, 1)$, the smallest $\bar{\sigma}_\rho^2$ at which $\mathbb{P}[\text{clip} \mid \pi] \geq 1 - \delta$ will be attained earlier for low-probability tokens.

This yields a regime in which $F_{\epsilon_h, \epsilon_l}(\sigma_\rho(\pi))$ is strongly suppressed on $I_L$ while remaining comparatively less suppressed on $I_H$, leading to $|\bar{G}_H| > |\bar{G}_L|$.

### B.3.5. CONCLUSION (EXISTENCE OF A REVERSAL THRESHOLD)

Combining the near on-policy inequality $|\bar{G}_L| > |\bar{G}_H|$ with the sufficiently off-policy inequality $|\bar{G}_H| > |\bar{G}_L|$ implies the existence of a threshold $\tau > 0$ such that the dominance reverses as $\bar{\sigma}_\rho^2$ crosses $\tau$. This matches Corollary 2.3.

## C. Geometric Direction Analysis with Principal Angles

This appendix details the geometric tool used in Section 3.1 to quantify the *directional* similarity between high-dimensional gradients. Our goal is to compare gradients in a way that is (i) robust to global rescaling, (ii) stable in very high dimensions, and (iii) compatible with the matrix-shaped structure of transformer layer gradients.

### C.1. Why subspace alignment instead of vector angles

A naive approach is to flatten all parameters and compute a cosine similarity between two gradient vectors. However, for large models this is often dominated by noise and by the many low-energy directions. In addition, gradients in neural networks are typically *low-rank structured*: most update energy concentrates in a relatively small set of directions (e.g., due to correlated features and shared activations). For these reasons, we compare the *dominant gradient subspaces* rather than individual flattened vectors.

Concretely, given a layer parameter matrix (e.g., a linear projection weight) with gradient

$$W \in \mathbb{R}^{m \times n},$$

we treat $W$ as a linear operator and extract its top-$k$ right singular vectors as a compact representation of its dominant update directions.

### C.2. Subspace representation via SVD

For each gradient matrix $W$, we compute the singular value decomposition (SVD):

$$W = U\Sigma V^\top,$$

where $U \in \mathbb{R}^{m \times m}$ and $V \in \mathbb{R}^{n \times n}$ are orthonormal matrices, and $\Sigma$ is diagonal with singular values in non-increasing order.

We define the rank-$k$ dominant *right* singular subspace

$$\mathcal{S}(W; k) = \mathrm{span}(V_k),$$

where $V_k \in \mathbb{R}^{n \times k}$ contains the top-$k$ columns of $V$. Intuitively, $V_k$ captures the principal directions in the input-feature space along which the gradient operator varies most strongly.

**Choice of right singular vectors.** We use right singular vectors because for weight matrices mapping $\mathbb{R}^n \to \mathbb{R}^m$, $V_k$ corresponds to directions in the input dimension. Empirically we found $V_k$ yields stable comparisons across layers of similar shapes. (Using left singular vectors $U_k$ gives qualitatively similar conclusions for our use case.)

**Choosing $k$.** We set $k = 128$ in all experiments unless otherwise noted. This value is large enough to capture most gradient energy for typical transformer projection matrices, while keeping computations tractable. In preliminary sweeps, we found the mean principal angle is qualitatively stable for $k$ in a broad range (e.g., 64–256), provided $k$ does not exceed the smaller matrix dimension.

### C.3. Principal angles between gradient subspaces

Given two gradients $W^A$ and $W^B$ for the same layer (e.g., computed under two different update constructions), we form orthonormal bases for their dominant subspaces:

$$Q^A \in \mathbb{R}^{n \times k}, \quad Q^B \in \mathbb{R}^{n \times k},$$

where $Q^A = V_k(W^A)$ and $Q^B = V_k(W^B)$.

The $k$ principal angles $\{\theta_i\}_{i=1}^k$ between the two subspaces are defined by:

$$\cos(\theta_i) = s_i,$$

where $\{s_i\}_{i=1}^k$ are the singular values of the cross-Gram matrix

$$M = (Q^A)^\top Q^B \in \mathbb{R}^{k \times k}.$$

Equivalently, if $M = \tilde{U} \, \mathrm{diag}(s_1, \ldots, s_k) \, \tilde{V}^\top$, then

$$\theta_i = \arccos(s_i), \quad i = 1, \ldots, k.$$

**Interpretation.** Each $\theta_i \in [0°, 90°]$ measures how well the $i$-th most-alignable direction in one subspace matches the other. In particular:

- $\theta_i = 0°$ indicates a perfectly shared direction.

- $\theta_i = 90°$ indicates orthogonality for that principal direction.

Small angles across many $i$ imply the two gradients share a similar dominant update subspace; large angles indicate a substantial rotation of dominant directions.

### C.4. Summary statistics reported

The set of principal angles yields a spectrum of directional similarity. For compact reporting, we aggregate angles into summary statistics.

**Mean principal angle.** We report the mean principal angle:

$$\bar{\theta}(W^A, W^B) \;=\; \frac{1}{k} \sum_{i=1}^{k} \theta_i.$$

This provides a single scalar notion of subspace alignment. In our plots/tables, smaller $\bar{\theta}$ indicates stronger alignment.

**Alternative aggregations (not required).** Other useful summaries include the median, the maximum angle, or a weighted mean using singular values of $W$ as weights. We found the unweighted mean sufficient and stable for the comparisons in this paper.

### C.5. Numerical stability

In exact arithmetic, all singular values of $M = (Q^A)^\top Q^B$ lie in $[0, 1]$. In finite precision, slight violations (e.g., $1 + 10^{-7}$) can occur, which would make $\arccos(\cdot)$ undefined. We therefore clamp:

$$s_i \leftarrow \min(1, \max(0, s_i)),$$

before applying $\arccos$.

### C.6. Layer-wise computation and aggregation across layers

We compute principal angles *per layer* and (when presenting model-level summaries) aggregate across layers.

**Per-layer procedure.** For each layer weight matrix:

1. Compute gradients $W^A$ and $W^B$ under two update constructions.

2. Compute truncated SVDs to obtain $Q^A$ and $Q^B$.

3. Form $M = (Q^A)^\top Q^B$, compute its singular values $\{s_i\}$, then angles $\{\theta_i\}$.

4. Summarize with $\bar{\theta}$ (and optionally the full spectrum).

**Across-layer aggregation.** To summarize directional similarity for a whole model, we average $\bar{\theta}$ over a set of layers. In the main experiments we include the main projection matrices (e.g., attention and MLP projections) and exclude bias vectors and layernorm parameters, for which matrix-shaped SVD-based comparisons are not meaningful.

### C.7. Practical notes

**Truncated SVD implementation.** We use a truncated SVD to compute the top-$k$ singular vectors efficiently. For large matrices, randomized SVD yields significant speedups while maintaining sufficient accuracy for principal-angle estimates.

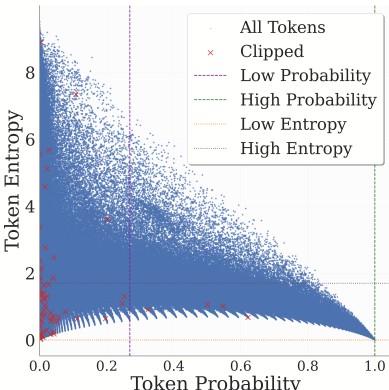

*Figure 7.* Correlation between token probability and entropy, with clipped tokens highlighted (red crosses).

**Why compare subspaces, not singular vectors directly.** Singular vectors are defined up to sign flips and may swap under small perturbations when singular values are close. Subspace-based measures are invariant to such instabilities, making principal angles more reliable for comparing gradients from stochastic minibatches.

### C.8. Limitations

Principal angles measure similarity of dominant *linear* subspaces per layer. This is not a full characterization of optimization trajectories, and it does not capture higher-order geometry or parameter coupling across layers. Nevertheless, it is a simple, robust diagnostic for whether two update constructions point in broadly similar directions in parameter space.

### C.9. Reproducibility checklist

For reproducibility, we specify:

- Subspace dimension: $k = 128$.

- Angle computation: SVD of $(Q^A)^\top Q^B$ with clamping to $[0, 1]$.

- Layer selection: matrix-shaped parameters (projection weights); exclude biases and layernorm scalars.

- Reported statistic: mean principal angle $\bar{\theta}$ (and spectra in some ablations).

## D. Reconciling Heuristics: Probability, Entropy, and Clipping Dynamics

The empirical contradiction is resolved by examining the interplay between probability, entropy, and the clipping mechanism. While a token's probability $\pi$ and the vocabulary's entropy $H$ are correlated, their formal relationship is bounded:

$$\underbrace{-\pi \log \pi - (1 - \pi) \log(1 - \pi)}_{H_{\min}(\pi)} \le H \le \underbrace{-\pi \log \pi - (1 - \pi) \log\left(\frac{1 - \pi}{V - 1}\right)}_{H_{\max}(\pi)}$$

Figure 7 visualizes these bounds and provides the crucial insight: clipped tokens (red crosses) are not randomly distributed but are overwhelmingly concentrated in the low-probability, low-entropy region.

This single observation directly reconciles the conflicting findings:

- Updating **low-probability tokens** (Yang et al., 2025b) is suboptimal because this population is dominated by dynamically unstable, low-entropy tokens. As established in Lemma 2.1, these tokens are highly susceptible to clipping, which attenuates their gradient contribution.

- Conversely, updating **high-entropy tokens** (Wang et al., 2025b) succeeds because it acts as an effective filter for stability. This heuristic implicitly avoids the heavily clipped region, selecting for tokens from more uniform distributions that are robust to policy updates.

Therefore, entropy is not merely a proxy for low probability but a more discerning indicator of a token's dynamic stability under the clipping pressures that govern off-policy optimization.

## E. Detailed Gradient Analysis

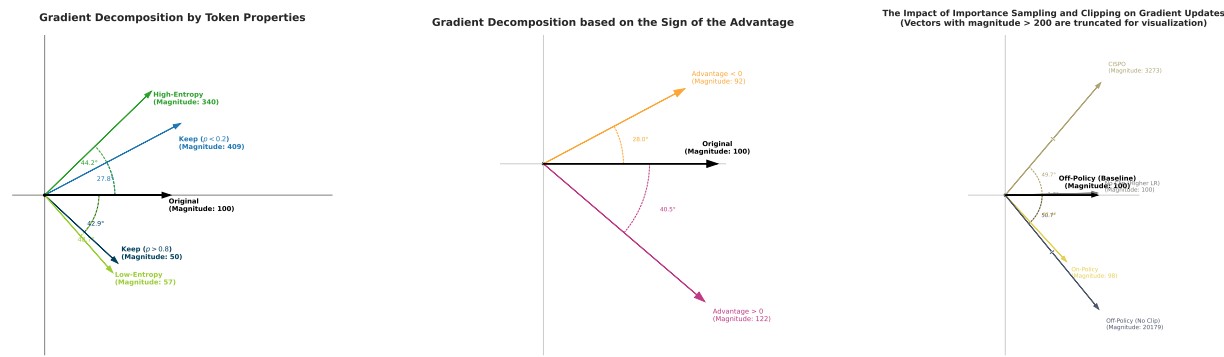

*Figure 8.* The impact of different key factors on RL updates.

### E.1. Token Property Analysis

To understand the fundamental composition of the policy gradient, we analyze the contributions of different token subsets based on their intrinsic properties. Our analysis, presented in Figure 8 left, isolates the gradient signals originating from distinct populations of tokens, revealing that the update is overwhelmingly dominated by a small, low-probability subset. Consistent with Yang et al. (2025b), tokens with $\pi_k < 0.2$ produce a gradient with a norm 409% that of the full gradient and are exceptionally well-aligned with its direction ($27.8°$ mean principal angle). In stark contrast, high-probability tokens ($\pi_k > 0.8$) contribute a much smaller, less aligned gradient (50% norm, $42.8°$ angle). This confirms that the policy gradient is primarily driven by powerful but potentially unstable signals from the tail of the probability distribution.

The strategy of updating high-entropy tokens, proposed by Wang et al. (2025b), can be understood as an effective heuristic for managing this instability. The high-entropy subset produces a gradient with a substantial magnitude (340% of original) but a weaker directional alignment ($44.2°$) than the pure low-probability set. This suggests the high-entropy criterion functions as a filter: it selects for tokens that are informative (often having low to moderate probability) but originate from flatter, more uncertain distributions. As we will show, these flatter distributions are inherently more stable under policy updates. Therefore, the heuristic succeeds by implicitly balancing gradient magnitude against stability. This success underscores the need for a more principled mechanism that can dynamically manage this trade-off.

### E.2. Advantage Sign Analysis

The advantage function, $A$, guides learning by signaling whether an action should be reinforced ($A > 0$) or suppressed ($A < 0$). To understand their distinct roles, we analyze the gradient contributions from these two subsets separately, as shown in Figure 8 middle. Our analysis reveals a critical asymmetry: while the gradient norms from positive (122% of original) and negative (92%) advantage samples are comparable, their directional guidance differs markedly. The gradient from negative-advantage samples is substantially better aligned with the total gradient direction ($28.0°$ mean principal angle) than that from positive-advantage samples ($40.5°$). This indicates that the overall update direction is predominantly dictated by corrective signals from suboptimal actions.

This directional dominance stems from the fundamentally different effects of positive and negative updates. An update with $A > 0$ reinforces a single action, implicitly suppressing all alternatives and potentially narrowing the policy. Conversely,

an update with $A < 0$ penalizes a specific action, which effectively redistributes probability mass across the rest of the vocabulary. This latter mechanism, a form of error correction, promotes exploration and enhances policy diversity. The geometric dominance of negative-advantage gradients therefore suggests that learning in complex tasks is driven more by correcting errors than by reinforcing known correct pathways, a process crucial for discovering robust strategies. While Zhu et al. (2025) also noted the importance of negative samples, our work provides a novel perspective by demonstrating their dominant role in shaping the gradient's geometric direction.

### E.3. Importance Sampling Analysis

To improve sample efficiency, off-policy reinforcement learning corrects for the policy distribution mismatch using Importance Sampling (IS). However, the high variance of the IS ratio necessitates a stabilization mechanism like clipping, which modulates the raw gradient into a stable update signal. To dissect this process, we compare our baseline off-policy GRPO update against several variants (Figure 8 right). The indispensability of clipping is starkly illustrated by its removal: the gradient norm explodes to **20,177%** of the baseline, and its direction severely deviates (**50.7°** mean principal angle). This confirms clipping is crucial not just for controlling magnitude but also for maintaining a stable optimization path.

Perhaps the most revealing finding is the significant directional divergence (**47.4°**) between the on-policy and standard off-policy (GRPO) gradients, despite their nearly identical magnitudes (98% vs. 100%). This highlights a fundamental trade-off: to maintain magnitude stability, GRPO's fixed clipping mechanism systematically filters out certain updates (predominantly from high-variance, low-probability tokens), creating a gradient direction that is geometrically distinct from the on-policy ideal. This implies that while off-policy learning is sample-efficient, its optimization trajectory can substantially differ from its on-policy counterpart.

The specific design of the clipping strategy further modulates this trade-off. For instance, CISPO, which clamps the IS ratio instead of gating the entire gradient to zero, retains more signal from outlier tokens. This results in a substantially larger gradient norm (**3,268%**) and a similarly large directional shift (**49.7°**), reflecting a more aggressive update policy. In contrast, a simple asymmetric clipping scheme ($\epsilon_h > \epsilon_l$), inspired by DAPO, showed negligible impact in our setting ($1.3°$ divergence). These comparisons reveal that the standard clipping in methods like PPO/GRPO is not a neutral stabilizer; it actively shapes the gradient by discarding certain information. This suggests an opportunity for adaptive mechanisms that can better preserve the on-policy direction while retaining off-policy efficiency.

## F. ACPO Formulation (Details)

ACPO replaces the single global clipping range in Eq. 1 with a bin-specific range determined by the behavior-token probability. Given an update mini-batch (pooling all tokens in the update batch), we assign each token $(i, t)$ to one of $B$ equal-width bins on $[0, 1]$ via

$$b(i, t) = \min\{B, \lfloor B \cdot \pi_{\theta_{\text{old}}}(o_{i,t} \mid q, o_{i,<t}) \rfloor + 1\}. \tag{51}$$

For each bin $b$, we compute the within-bin dispersion of IS ratios on this mini-batch and set the clipping range as

$$\sigma_b = \text{Std}\big(\{\rho_{i,t} : b(i, t) = b\}\big),$$
$$\epsilon_b = \epsilon_{\text{base}} + \alpha \, \sigma_b. \tag{52}$$

We then apply token-wise clipping by using $\epsilon_{b(i,t)}$ in Eq. 1. The resulting objective is

$$J_{\text{ACPO}}(\theta) = \mathbb{E}_{q \sim P(Q), \{o_i\}_{i=1}^G \sim \pi_{\theta_{\text{old}}}(O|q)} \frac{1}{G} \sum_{i=1}^{G} \frac{1}{|o_i|} \sum_{t=1}^{|o_i|}$$
$$\min\Big[\rho_{i,t} A_i, \ \text{clip}\big(\rho_{i,t}, 1 - \epsilon_{b(i,t)}, 1 + \epsilon_{b(i,t)}\big) A_i\Big]. \tag{53}$$

## G. More Related Works

**Reinforcement Learning for Large Language Models.** Reinforcement learning has evolved from a tool for preference alignment to a key technique for enhancing reasoning capabilities in large language models. Initially pioneered through Reinforcement Learning from Human Feedback (RLHF), RL methods like PPO (Schulman et al., 2017) were used to align models with human preferences using human-annotated data (Ouyang et al., 2022). The landscape shifted dramatically with

---

**Algorithm 1** ACPO (bin-wise adaptive clipping)

---

1: **Input:** behavior policy $\pi_{\theta_{\text{old}}}$, current policy $\pi_\theta$, batch of rollouts, $B$, $\epsilon_{\text{base}}$, $\alpha$.

2: Compute token-wise IS ratios $\rho_{i,t} = \frac{\pi_\theta(o_{i,t}|q,o_{i,<t})}{\pi_{\theta_{\text{old}}}(o_{i,t}|q,o_{i,<t})}$.

3: Assign each token $(i,t)$ to a probability bin $b(i,t) = \min\{B, \lfloor B \cdot \pi_{\theta_{\text{old}}}(o_{i,t} \mid q, o_{i,<t})\rfloor + 1\}$.

4: **for** $b = 1, \ldots, B$ **do**

5: $\quad \sigma_b \leftarrow \text{Std}(\{\rho_{i,t} : b(i,t) = b\})$.

6: $\quad \epsilon_b \leftarrow \epsilon_{\text{base}} + \alpha\, \sigma_b$.

7: **end for**

8: Optimize Eq. 53 using token-wise clipping bounds $(1 - \epsilon_{b(i,t)},\, 1 + \epsilon_{b(i,t)})$.

---

the emergence of RL with verifiable rewards (RLVR), which leverages objective, automatically verifiable feedback signals instead of subjective human preferences (Lambert et al., 2024). OpenAI's o1 model (Jaech et al., 2024) first showcased that RLVR can effectively incentivize reasoning at scale, particularly in tasks like mathematics and programming. Building on this foundation, researchers developed improved algorithmic methods such as GRPO (Shao et al., 2024) and its variants (e.g., DAPO (Yu et al., 2025)). Subsequent models trained with these methods, including DeepSeek-R1 (Guo et al., 2025a), QwQ (Team, 2025), and AceReason-Nemotron (Chen et al., 2025c), demonstrate that strong reasoning capabilities can emerge through outcome-based optimization with online RL algorithms, establishing RLVR as a promising paradigm for developing reasoning-capable LLMs.

**Token-Level Dynamics and Update Strategies.** While RLVR has demonstrated strong potential, its effectiveness is often determined by how token-level updates are applied during training. Recent research has explored multiple related directions : entropy- or probability-based methods that incorporate or reduce uncertainty (Wang et al., 2025b; Yang et al., 2025b; Cheng et al., 2025; Cui et al., 2025; Gao et al., 2025; Chen et al., 2025b), modifications to importance-sampling and clipping that stabilize gradients (Roux et al., 2025; Chen et al., 2025a; Su et al., 2025; Zheng et al., 2025a; Wang et al., 2025a), advantage design strategies ranging from negative reinforcement to minimalist or segment-level credit assignment (Zhu et al., 2025; Xiong et al., 2025), and off-policy approaches that reshape update distributions (Yan et al., 2025; Ma et al., 2025; Fu et al., 2025; Arnal et al., 2025). However, studies in this area sometimes yield conflicting conclusions: for example, Wang et al. (2025b) emphasize that high-entropy minority tokens drive effective learning, whereas Yang et al. (2025b) show that low-probability(usually high-entropy) tokens can over-dominate gradients and should be suppressed. Such inconsistencies calls for a deeper, component-wise analysis of the RL update pipeline. While recent work has started to examine how individual training techniques affect RL dynamics (Liu et al., 2025), we still lack a comprehensive framework that can systematically explain how these components work together to drive effective policy updates in LLM reasoning.

# H. Training Details

We conduct experiments using the Qwen2.5-7B model on the orz dataset. All methods are evaluated under two settings: near on-policy (N-OnP) with a train batch size of 128 and PPO mini batch size of 64, and off-policy (OffP) with a train batch size of 256 and PPO mini batch size of 16. The key difference between these settings lies in the staleness of the rollout data: in the N-OnP setting, the policy is updated more frequently relative to the rollout data, while in the OffP setting, the policy undergoes more gradient steps before refreshing the rollout buffer.

All methods share the following common hyperparameters: a learning rate of 1e-6, a maximum prompt length of 1024 tokens, a maximum response length of 3072 tokens, and a rollout size $N = 10$. We train for 15 epochs without KL regularization (KL coefficient = 0) and without entropy bonus (entropy coefficient = 0). The loss aggregation strategy is seq-mean-token-mean, which first averages over tokens within each sequence and then averages across sequences.

**DAPO.** The baseline method uses asymmetric clipping with $\epsilon_{\text{low}} = 0.2$ and $\epsilon_{\text{high}} = 0.3$, without any additional modifications.

**AR-Lopti.** This method combines Advantage Reweighting (AR) and Low-Probability Token Isolation (Lopti). The AR component reweights advantages using $\alpha_{\text{AR}} = 0.3$, $\tau_{\text{AR}} = 0.7$, and negative advantage weight $w_{\text{neg}} = 1.0$. The Lopti component isolates low-probability tokens using a threshold $p_{\text{lopti}} = 0.5$.

*Table 3.* Common hyperparameters for Qwen2.5-7B experiments on MATH.

| Hyperparameter | N-OnP | OffP |
|---|---|---|
| Base Model | Qwen2.5-7B | |
| Max Prompt Length | 1024 | |
| Max Response Length | 3072 | |
| Learning Rate | 1e-6 | |
| Total Epochs | 15 | |
| Rollout $N$ | 10 | |
| Temperature | 1.0 | |
| Validation $N$ | 8 | |
| $\epsilon_{\text{low}}$ | 0.2 | |
| $\epsilon_{\text{high}}$ | 0.3 | |
| Entropy Coefficient | 0.0 | |
| KL Coefficient | 0.0 | |
| Loss Aggregation | seq-mean-token-mean | |
| Train Batch Size | 128 | 256 |
| PPO Mini Batch Size | 64 | 16 |

**High-Entropy.** This method selectively updates only the tokens with the highest entropy. We set $k_{\text{ratio}} = 0.2$, meaning only the top 20% highest-entropy tokens are updated during training.

**Low-Entropy.** Conversely, this method updates only the tokens with the lowest entropy. We set $k_{\text{ratio}} = 0.8$, meaning only the bottom 80% (i.e., lowest-entropy) tokens are updated.

**ACPO.** Adaptive Clipping Policy Optimization dynamically adjusts the clipping range based on advantage variance. We use $\alpha = 3.0$ and $\epsilon_{\text{base}} = 0.2$, with the adaptive clip range bounded between $\epsilon_{\text{min}} = 0.0$ and $\epsilon_{\text{max}} = 3.0$.

**CISPO.** This method applies importance sampling-aware clipping with separate clip ratios for the importance sampling ratio: $\epsilon_{\text{is\_high}} = 0.45$ and $\epsilon_{\text{is\_low}} = 1.0$.

# I. Ablation Study

## I.1. Parameter Sensitivity

We study the sensitivity of ACPO to its two main hyperparameters: the scaling factor $\alpha$ in the adaptive rule $\epsilon_b = \epsilon_{\text{base}} + \alpha \, \sigma_b$, and the base clipping range $\epsilon_{\text{base}}$. Table 5 reports results on ORZ-trained models, evaluated on in-domain and OOD math benchmarks, under both near on-policy (2 updates/rollout) and off-policy (16 updates/rollout) regimes.

Overall, ACPO is reasonably robust across a broad range of settings. In particular, moderate values (e.g., $\alpha \in \{3, 4\}$ with $\epsilon_{\text{base}} \approx 0.2$) achieve consistently strong average performance, while the per-benchmark optima can shift slightly between the on-policy and off-policy regimes, reflecting different clipping dynamics.

## I.2. Effects of Clipping Thresholds

A natural question is whether ACPO's improvements stem from the adaptive mechanism itself, or simply from using different clipping threshold values. To disentangle these factors, we conduct ablation experiments on the Countdown task with Qwen2.5-3B-Instruct (Table 6), where we match fixed thresholds to ACPO's observed statistics: (1) **GRPO (Clip=Avg)**: both upper and lower thresholds set to ACPO's weighted average clip range; (2) **DAPO (Clip-High=Max)**: upper threshold set to ACPO's maximum observed clip range.

Under the *avg* setting, the uniform threshold is dominated by high-probability tokens with low variance, making it more restrictive than standard GRPO. This overly tight bound suppresses valuable updates from low-probability tokens, reducing performance (72.55% vs. 73.27% baseline). Under the *max* setting, matching DAPO's upper bound to ACPO's maximum

*Table 4.* Method-specific hyperparameters.

| Method | Specific Hyperparameters |
|---|---|
| DAPO | (Baseline, no additional parameters) |
| AR-Lopti | $\alpha_{\text{AR}} = 0.3$ 
 $\tau_{\text{AR}} = 0.7$ 
 $w_{\text{neg}} = 1.0$ 
 $p_{\text{lopti}} = 0.5$ |
| High-Entropy | entropy_top_k = True 
 $k_{\text{ratio}} = 0.2$ |
| Low-Entropy | entropy_bottom_k = True 
 $k_{\text{ratio}} = 0.8$ |
| ACPO | variance_adaptive_clip = True 
 $\alpha = 3.0$ 
 $\epsilon_{\text{base}} = 0.2$ 
 $\epsilon_{\text{min}} = 0.0$ 
 $\epsilon_{\text{max}} = 3.0$ |
| CISPO | $\epsilon_{\text{is\_high}} = 0.45$ 
 $\epsilon_{\text{is\_low}} = 1.0$ |

admits more signals and yields modest gains (74.65% vs. 73.27%). However, this uniform expansion ignores token-level variance: some tokens remain over-clipped while others become overly permissive, leaving performance below ACPO (75.74%).

These results confirm that ACPO's benefit stems from adaptive calibration, rather than simply using larger or smaller threshold values.

## J. Computational Overhead Analysis

We analyze the computational overhead of different methods by measuring the per-token update time during training. Table 7 summarizes the results across both Off-Policy and Near On-Policy settings.

**Why ACPO has negligible overhead.**  ACPO (Adaptive Clipping Policy Optimization) adapts the clipping threshold based on the variance of advantages within each batch. The additional computation involves: (1) computing the standard deviation of advantages (a single reduction operation), and (2) adjusting the clip ratio accordingly. Both operations are simple scalar computations performed once per batch, adding negligible overhead to the per-token update time.

**Why AR-Lopti has significant overhead.**  AR-Lopti combines two techniques that both require additional per-token computations:

- **Lopti (Low-Probability Token Isolation)**: Requires computing the probability of each token, comparing against a threshold ($p_{\text{lopti}} = 0.5$), and creating a binary mask to isolate low-probability tokens. This involves additional tensor operations and conditional logic.

- **AR (Advantage Reweighting)**: Applies a non-linear reweighting function to advantages based on their signs and magnitudes, which requires additional computation beyond simple clipping.

Together, these operations result in over $2\times$ computational overhead compared to DAPO.

**Why High-Entropy has moderate overhead.**  The High-Entropy method requires computing the entropy of each token's output distribution, sorting tokens by entropy, and selectively updating only the top-$k$ highest-entropy tokens. The entropy computation and sorting operations add approximately 40% overhead.

*Table 5.* Parameter sensitivity of ACPO on ORZ-trained models, evaluated on multiple mathematical reasoning benchmarks. Within each regime, the best score for each benchmark is **bolded**.

| $\alpha$ | $\epsilon_{base}$ | Minerva | Math500 | AMC | AIME24 | AIME25 | Olympiad | AVG |
|---|---|---|---|---|---|---|---|---|
| \multicolumn{9}{c}{**Near On-Policy (2 updates/rollout)**} |
| 2 | 0.0 | 29.97 | 77.85 | 47.84 | 19.02 | 12.87 | 41.06 | 38.10 |
| 2 | 0.1 | 30.55 | 78.77 | 50.78 | 19.51 | 12.70 | 41.80 | 39.02 |
| 2 | 0.2 | **31.12** | 79.68 | 53.72 | 20.00 | 12.53 | 42.53 | 39.93 |
| 2 | 0.3 | 30.83 | 79.13 | 52.54 | 20.18 | 12.47 | 42.69 | 39.64 |
| 3 | 0.0 | 29.87 | 77.95 | 48.04 | 19.58 | 13.75 | 41.20 | 38.40 |
| 3 | 0.1 | 30.45 | 78.87 | 50.98 | 20.31 | 13.50 | 42.10 | 39.37 |
| 3 | 0.2 | 31.02 | **79.78** | **53.92** | 20.80 | 13.33 | 42.83 | **40.28** |
| 3 | 0.3 | 30.73 | 79.23 | 52.74 | 20.98 | 13.27 | 42.99 | 39.99 |
| 4 | 0.0 | 29.47 | 77.65 | 47.24 | 20.32 | **14.17** | 41.56 | 38.40 |
| 4 | 0.1 | 30.05 | 78.56 | 50.18 | 20.81 | 14.00 | 42.30 | 39.32 |
| 4 | 0.2 | 30.62 | 79.48 | 53.12 | 21.30 | 13.83 | 43.03 | 40.23 |
| 4 | 0.3 | 30.33 | 78.93 | 51.94 | **21.48** | 13.77 | **43.19** | 39.94 |
| \multicolumn{9}{c}{**Off-Policy (16 updates/rollout)**} |
| 2 | 0.0 | 29.30 | 74.90 | 45.48 | 14.71 | 9.05 | 39.99 | 35.57 |
| 2 | 0.1 | 30.38 | 77.59 | 48.19 | 14.88 | 9.55 | 40.83 | 36.90 |
| 2 | 0.2 | 31.46 | 80.28 | 50.90 | 15.05 | 10.05 | 41.67 | 38.24 |
| 2 | 0.3 | 30.92 | 78.67 | 49.82 | 15.13 | 10.30 | 41.86 | 37.78 |
| 3 | 0.0 | 29.50 | 75.10 | 45.78 | 15.83 | 10.00 | 40.30 | 36.09 |
| 3 | 0.1 | 30.58 | 77.79 | 48.49 | 16.08 | 10.75 | 41.33 | 37.50 |
| 3 | 0.2 | **31.66** | **80.48** | **51.20** | 16.25 | 11.25 | 42.17 | **38.84** |
| 3 | 0.3 | 31.12 | 78.87 | 50.12 | 16.33 | 11.50 | 42.36 | 38.38 |
| 4 | 0.0 | 29.35 | 75.00 | 45.28 | 16.21 | 10.55 | 40.64 | 36.17 |
| 4 | 0.1 | 30.43 | 77.69 | 47.99 | 16.38 | 11.05 | 41.48 | 37.50 |
| 4 | 0.2 | 31.51 | 80.38 | 50.70 | 16.55 | 11.55 | 42.32 | **38.84** |
| 4 | 0.3 | 30.97 | 78.77 | 49.62 | **16.63** | **11.80** | 42.51 | 38.38 |

In summary, ACPO achieves its performance improvements through a computationally efficient advantage transformation, making it practical for large-scale training without additional infrastructure requirements.

## K. Statistical Significance Tests

We conduct statistical significance tests to compare ACPO against baseline methods on the OOD math benchmarks. For each benchmark, we perform independent two-sample Welch's $t$-tests (unequal variances) using a one-tailed alternative hypothesis that ACPO outperforms the baseline: $H_1 : \mu_{ACPO} > \mu_{baseline}$.

The test statistic is computed as:

$$t = \frac{\bar{X}_{ACPO} - \bar{X}_{baseline}}{\sqrt{\frac{s_{ACPO}^2}{n_{ACPO}} + \frac{s_{baseline}^2}{n_{baseline}}}}, \tag{54}$$

where $\bar{X}$ denotes the sample mean, $s$ the sample standard deviation, and $n$ the sample size. The degrees of freedom are approximated using the Welch–Satterthwaite equation. We use $n = 8$ for Minerva, Math500, AMC, and Olympiad benchmarks, and $n = 32$ for AIME24 and AIME25. We consider results statistically significant when $p < 0.05$.

*Table 6.* Ablation study on Countdown with Qwen2.5-3B-Instruct.

| Method / Setting | N-OnP. | OffP. |
|---|---|---|
| DAPO (Baseline) | 73.27 | 74.38 |
| DAPO (Clip-High=Max) | 74.65 | 75.86 |
| GRPO (Clip=Avg) | 72.55 | 74.16 |
| **ACPO** | **75.74** | **76.27** |

*Table 7.* Update time comparison. We report both per-step update time (seconds) and per-token update time (milliseconds). ACPO introduces negligible computational overhead compared to DAPO, while AR-Lopti incurs significant overhead due to additional token-level operations.

| Method | OffP | | | | N-OnP | | | |
|---|---|---|---|---|---|---|---|---|
| | Update (s) | Rel. | Per-token (ms) | Rel. | Update (s) | Rel. | Per-token (ms) | Rel. |
| DAPO | $231.0 \pm 31.7$ | – | $0.110 \pm 0.005$ | – | $440.1 \pm 48.9$ | – | $0.118 \pm 0.003$ | – |
| CISPO | $239.6 \pm 24.6$ | +3.8% | $0.128 \pm 0.006$ | +15.9% | $485.1 \pm 49.1$ | +10.2% | $0.136 \pm 0.004$ | +14.7% |
| AR-Lopti | $359.4 \pm 32.8$ | +55.6% | $0.234 \pm 0.006$ | +112.3% | $659.9 \pm 72.5$ | +49.9% | $0.258 \pm 0.017$ | +117.7% |
| High-Entropy | $181.9 \pm 22.9$ | −21.2% | $0.114 \pm 0.005$ | +3.3% | $556.1 \pm 107.6$ | +26.4% | $0.167 \pm 0.028$ | +40.6% |
| **ACPO (Ours)** | $219.1 \pm 45.5$ | −5.2% | $0.110 \pm 0.005$ | +0.0% | $458.4 \pm 84.6$ | +4.2% | $0.109 \pm 0.005$ | −8.0% |

*Table 8.* One-tailed Welch's $t$-test $p$-values comparing ACPO against each baseline on OOD math benchmarks, under near on-policy (2 updates/rollout) and off-policy (16 updates/rollout) settings. Bold indicates $p < 0.05$.

| **Near On-Policy (2 updates/rollout)** | | | | | | |
|---|---|---|---|---|---|---|
| ACPO vs. | Minerva | Math500 | AMC | AIME24 | AIME25 | Olympiad |
| Base | < 0.001 | < 0.001 | < 0.001 | < 0.001 | < 0.001 | < 0.001 |
| AR-Lopti | **0.017** | < 0.001 | < 0.001 | 0.052 | < 0.001 | < 0.001 |
| Top-20 | **0.018** | < 0.001 | **0.029** | **0.002** | < 0.001 | < 0.001 |
| Bottom-80 | **0.007** | < 0.001 | < 0.001 | 0.209 | < 0.001 | < 0.001 |
| DAPO | 0.143 | **0.002** | **0.007** | – | **0.005** | **0.007** |
| CISPO | 0.285 | **0.008** | < 0.001 | – | < 0.001 | < 0.001 |
| **Off-Policy (16 updates/rollout)** | | | | | | |
| ACPO vs. | Minerva | Math500 | AMC | AIME24 | AIME25 | Olympiad |
| Base | < 0.001 | < 0.001 | < 0.001 | < 0.001 | < 0.001 | < 0.001 |
| AR-Lopti | < 0.001 | < 0.001 | < 0.001 | < 0.001 | < 0.001 | < 0.001 |
| Top-20 | **0.005** | < 0.001 | < 0.001 | **0.013** | – | < 0.001 |
| Bottom-80 | < 0.001 | < 0.001 | < 0.001 | < 0.001 | < 0.001 | < 0.001 |
| DAPO | **0.027** | < 0.001 | **0.001** | – | < 0.001 | < 0.001 |
| CISPO | **0.007** | < 0.001 | **0.003** | **0.002** | **0.02** | < 0.001 |

