# OpenReview forum: "What are Key Factors for Updates in RL for LLM Reasoning?"
_ICML.cc/2026/Conference — Submitted to ICML 2026_

### Official Review · Reviewer_uYkU · 2026-02-25

**Soundness:** 3
**Presentation:** 3
**Significance:** 3
**Originality:** 3
**Overall Recommendation:** 4
**Confidence:** 3

**Summary:**

This paper identifies off-policy degree as the root cause of contradictory heuristics in RLVR. It reveals a gradient dominance reversal phenomenon: low-probability tokens dominate updates in near-on-policy settings, but high-probability tokens take over as off-policy drift increases due to excessive clipping. The proposed ACPO resolves this by replacing fixed clipping with adaptive, variance-aware thresholds per token-probability bin, ensuring more stable and effective policy updates.

**Compliance With Llm Reviewing Policy:**

Affirmed.

**Final Justification:**

I appreciate the authors’ response, which has largely addressed my previous concerns. I recommend to weak accept.

**Key Questions For Authors:**

1) If the distributional assumptions for $\rho$ and $A$ do not hold, the justification for variance-based thresholds (using only $\sigma_{\rho}$) weakens.
2)  It needs to be clearer that the benefit comes from adaptive clipping itself (not from removing baseline “tricks” in a controlled setup), and ideally that it improves optimization “correctness,” not just stability.

Please address the questions raised above. I will adjust the score accordingly after considering the authors’ responses during the rebuttal phase, as well as feedback from other reviewers.

**Limitations:**

1) In extreme heavy-tailed regimes, standard deviation alone may be an unstable summary statistic.
2) The paper lacks strong evidence (e.g., entropy curves) showing how adaptive clipping changes implicit exploration during training.

**Strengths And Weaknesses:**

Strengths.
1) Attributes the “high-entropy vs. high-probability token” discrepancy to off-policy degree and ties it to a clear gradient-dominance reversal story.
2) ACPO is a light modification to PPO/GRPO, binning tokens and adjusting clipping via simple batch statistics, so the added overhead is minimal.
3) Tested across ORZ-57K, HiTab, and Countdown, and shown to work on both 3B and 7B models.

Weaknesses.
1) Strong theoretical assumptions: Treating the IS ratio  $\rho$ and advantage $A$ as Gaussian and independent is hard to justify. $\rho$ is often heavy-tailed and $A$ can be skewed in LLM RL.
2) Logit-level analysis: The theory focuses on gradients w.r.t. logits rather than parameter gradients, which are more directly tied to the actual update magnitude and direction.
3) Marginal gains on hard benchmarks: Improvements over strong baselines (e.g., DAPO) can look small on tougher tasks, making the contribution feel incremental.

---

> ### Author Rebuttal · Authors · 2026-03-30
>
> Thank you for the detailed and constructive review. We address each concern below.
>
> > Q1 (W1 / KQ1): Theoretical assumptions and variance-threshold justification
>
> A1: The Gaussian and independence assumptions are simplifications. Their role is to support Proposition 1's closed-form derivation; Lemma 1—which establishes that lower-probability tokens have higher conditional IS-ratio variance—does not rely on them. Regarding whether variance-based thresholds remain justified: we believe they do. ACPO does not estimate the distribution of ρ or A from any parametric model; it computes the within-bin IS-ratio standard deviation directly from observed tokens in each update batch. The method's motivation is empirical: as illustrated in Figure 2, fixed clipping induces systematically uneven clipping rates across probability bins, and ACPO corrects this by setting per-bin thresholds proportional to locally observed IS-ratio spread. This mismatch and its correction do not require Gaussian or independence assumptions to hold exactly.
>
> We agree that in extreme heavy-tailed regimes, standard deviation may be less robust than alternatives. We will note this limitation in the revision. In the camera-ready appendix we will clarify the dependency structure: which conclusions rely on simplifying assumptions and which are structural results, and how the expressions change under alternative distributional assumptions.
>
> > Q2 (W2): Logit-level analysis
>
> A2: The final update is indeed a parameter-space update. Our use of logit-level analysis is therefore not to claim that it fully characterizes all parameter-space dynamics, but to study the mechanism most directly affected by clipping. In policy gradient, clipping changes the token-wise coefficient multiplying the update signal, and in logit space this appears as the source signal of backpropagation. Analyzing this source signal lets us isolate how clipping, advantage, and token probability determine whether a token's contribution is preserved, attenuated, or suppressed. This choice is also consistent with prior probability-/ratio-level analysis: PPO derives its surrogate objective and clipping mechanism in action-probability ratio space rather than parameter space, and the two RLVR works we aim to reconcile (Yang et al.; Wang et al.) also analyze token-level behavior through probability/entropy-based quantities. This makes logit space a natural level for isolating the clipping mechanism. We will clarify this scope in the revision.
>
> > Q3 (W3 / KQ2 part 1): Absolute gains and optimization correctness
>
> A3: Stability alone is not sufficient; what matters is better policy performance. Appendix K Welch's t-tests already show that ACPO significantly outperforms DAPO on multiple benchmarks, including Math500, AMC, AIME25, and Olympiad (p < 0.05). The absolute gains are also meaningful: on the ORZ/math task (7B), ACPO vs. DAPO improves Math500 off-policy from 76.72% to 80.48% (+3.76%), AMC off-policy from 46.84% to 51.20% (+4.36%), and Olympiad off-policy from 37.20% to 42.17% (+4.97%); on HiTab (3B, off-policy), it improves from 44.00% to 46.42% (+2.42%).
>
> AIME24 is a clear exception where DAPO performs better. Our point is therefore not that ACPO wins every benchmark, but that it improves optimization quality consistently in the regime where adaptive clipping is most needed—especially under off-policy training, where fixed clipping most strongly distorts which updates survive.
>
> > Q4 (KQ2 part 2): Benefit from adaptive clipping itself and controlled setup
>
> A4: In our experiments, all methods share an identical base configuration (learning rate, batch size, rollout size, KL coefficient, etc.), and ACPO changes only the clipping parameterization. This is stated in Section 5 and Appendix J.
>
> As for baseline tricks, DAPO includes components beyond clip-higher; in our paper we only apply its clip-higher component and do not enable additional tricks such as dynamic sampling. In our zero-KL GRPO setting, filtering all-0/all-1 reward samples mainly affects sample efficiency, since such samples contribute exactly zero gradient.
>
> More directly, we include a controlled threshold ablation in the appendix to isolate whether the gain comes from adaptive clipping itself or simply from changing threshold values. On Countdown (Qwen2.5-3B-Instruct), DAPO (Clip-High=Max) improves over the DAPO baseline (74.65/75.86 vs. 73.27/74.38), but still remains below ACPO (75.74/76.27). Conversely, GRPO (Clip=Avg) degrades to 72.55/74.16. This shows that the benefit comes from adaptive per-bin calibration itself, rather than from uniformly using a larger or smaller threshold.
>
> > Q5 (L2): Entropy curves and training dynamics
>
> A5: Thank you for this suggestion. We agree that entropy curves would help make the training dynamics more explicit, and we will include them in the revision.
>
> We would be happy to continue the discussion if any further clarification would be helpful.

---

> > ### Author Rebuttal · Reviewer_uYkU · 2026-04-01
> >
> > The author's rebuttal has largely resolved my confusion. Although there are some theoretical assumptions, I believe it has contributed to the community. I will improve my score. Good Luck!

---

> > > ### Author Response · Authors · 2026-04-01
> > >
> > > We sincerely appreciate your encouraging follow-up and are very glad to hear that our rebuttal helped clarify the paper. Thank you as well for recognizing the value of the contribution.

---

### Official Review · Reviewer_hSLS · 2026-02-26

**Soundness:** 2
**Presentation:** 2
**Significance:** 2
**Originality:** 2
**Overall Recommendation:** 4
**Confidence:** 3

**Summary:**

The authors investigate a paradox that whether focusing on high-entropy tokens help LLM or not during RL post training. They provide a theoretical analysis of how low- and high-probability tokens behave under varying degrees of off-policy training. Leveraging these insights, the authors introduce ACPO, a method that dynamically adjusts clipping boundaries based on the empirical variance of the importance sampling ratio. Empirical results over multiple datasets are conducted to support the proposed method.

**Compliance With Llm Reviewing Policy:**

Affirmed.

**Final Justification:**

The authors’ rebuttal has addressed my primary concerns. Adding the promised discussions and clarifying the scope in the final version will better position this work in the research field.

**Key Questions For Authors:**

See Weakness above.

**Limitations:**

The authors don't discuss the limitations.

**Strengths And Weaknesses:**

S1. The gradient dominance reversal is interesting.

S2.  The proposed method is motivated by a theoretical analysis of the previous empirical contradictions.

W1. While the theoretical framework is conceptually interesting, several assumptions appear to limit its practical utility. The analysis is conducted primarily on logits rather than model parameters, which may not fully capture the optimization dynamics of LLMs. The assumption that the importance sampling ratio and advantage follow a normal distribution—and are independent of one another—is a strong simplification that often does not hold in real-world settings.

W2. The proposed ACPO appears to be an incremental evolution of DAPO rather than a fundamental breakthrough. A key limitation is that ACPO does not eliminate the need for manual hyperparameter tuning, which limits its "adaptive" value. The empirical improvements over DAPO are marginal. Notably, on the AIME 24 benchmark, ACPO performs significantly worse than DAPO, which calls into question the robustness of the proposed method across different reasoning tasks.

W3. ACPO relies on bins to calculate variance. How sensitive the model is to the number of bins?

---

> ### Author Rebuttal · Authors · 2026-03-30
>
> Thank you for the constructive feedback.
>
> > Q1.1 (W1): Why analyze logits rather than parameters?
>
> A1.1: It is true that the final update is a parameter-space update. We use logit-level analysis not to fully characterize parameter-space dynamics, but to study the mechanism most directly affected by clipping. In policy gradient, clipping changes the token-wise coefficient multiplying the update signal, and in logit space this appears as the source signal of backpropagation. This lets us isolate how clipping, advantage, and token probability determine whether a token's contribution is preserved or suppressed. This choice is also consistent with prior probability-/ratio-level analysis: PPO derives clipping in action-probability ratio space, and the two RLVR works we aim to reconcile (Yang et al.; Wang et al.) also analyze token-level behavior through probability/entropy-based quantities, making logit space a natural level for isolating this mechanism. We will clarify this scope in the revision.
>
> > Q1.2 (W1/L): Are the Gaussian / independence assumptions too strong?
>
> A1.2: These are simplifying assumptions. Their main role is to yield a clean closed-form expression, rather than to make the core mechanism hold. Importantly, Lemma 1 does not rely on them: the structural result is that lower-probability tokens have higher conditional IS-ratio variance, and clipping suppresses these higher-variance tokens first. Figure 2 further verifies this structural trend empirically: lower-probability tokens exhibit larger IS-ratio spread and are more likely to be clipped.
>
> If Gaussianity is relaxed, the current truncated-normal form in terms of $\Phi$ and $\phi$ is replaced by the corresponding truncated-moment expression for the chosen distribution. If independence is relaxed, the factorized form becomes a joint truncated expectation. These changes affect compactness of the closed form, not the underlying mechanism. As long as (i) low-probability tokens have higher conditional IS-ratio variance, and (ii) clipping suppression increases with this variance, the qualitative conclusion, including the dominance reversal argument, remains unchanged. We will clarify this dependency structure, discuss alternative assumptions, and add a limitations discussion on these assumptions and the scope to clipped token-level IS in the camera-ready appendix.
>
> > Q2.1 (W2): ACPO seems incremental relative to DAPO.
>
> A2.1: While ACPO is simple, it addresses a key issue: existing methods use uniform clipping despite heterogeneous IS-ratio distributions across tokens. Our analysis shows that this mismatch becomes more problematic under off-policy training, and ACPO provides a targeted, mechanism-derived correction by adapting clipping boundaries based on within-bin IS-ratio dispersion. More broadly, the contribution is an analysis-to-design pipeline: identifying the mechanism explains prior empirical inconsistencies and directly yields an effective algorithm.
>
> > Q2.2 (W2): If ACPO still has hyperparameters, in what sense is it adaptive?
>
> A2.2: Here, “adaptive” means the clipping range is adjusted online according to observed within-bin IS-ratio statistics, rather than fixed globally. It does not mean hyperparameter-free. The hyperparameters are few and intuitive, and Appendix Table 5 shows that performance is stable across a broad range.
>
> > Q2.3 (W2): The gains over DAPO are marginal, and ACPO is worse on AIME24.
>
> A2.3: Appendix K provides Welch's t-tests showing statistically significant gains over DAPO on multiple math benchmarks (e.g., Math500, AMC, AIME25, and Olympiad; p < 0.05). The off-policy gains are also meaningful, including +3.76% on Math500, +4.36% on AMC, and +4.97% on Olympiad.
>
> AIME24 is a benchmark-specific exception where DAPO performs better. Our claim is not per-benchmark dominance, but consistent improvements across tasks and across both near on-policy and off-policy settings. From a broader view, ACPO achieves the best average rank (1.33) across 18 task-regime settings, with 12/18 first-place and 5/18 second-place finishes, and shows the smallest degradation from near on-policy to off-policy training (0.8%).
>
> > Q3 (W3): How sensitive is ACPO to the number of bins?
>
> A3: We add a bin-sensitivity ablation and find that ACPO is robust to moderate changes in bin number.
>
> |Regime|Bins|Minerva|Math500|AMC|AIME24|AIME25|Olympiad|Avg.|
> |---|---:|---:|---:|---:|---:|---:|---:|---:|
> |N-OnP.|4|30.93|78.60|52.56|19.58|12.08|41.86|39.27|
> ||5|31.02|79.78|54.37|16.77|14.37|42.83|39.86|
> ||6|30.58|78.88|53.92|17.92|14.17|41.33|39.47|
> |OffP.|4|30.65|80.28|50.75|17.08|10.42|40.64|38.30|
> ||5|31.66|80.48|51.20|16.35|10.10|42.17|38.66|
> ||6|30.10|79.48|51.81|16.67|9.17|42.43|38.28|
>
> |Bins|Countdown (N-OnP.)|Countdown (OffP.)|
> |---|---:|---:|
> |4|75.20|75.90|
> |5|75.74|76.27|
> |6|75.43|75.65|
>
> Paper's B=5 performs best, while B=4 and B=6 remain very close in all settings. We will include this ablation in the revision.
>
> We welcome further discussion.

---

> > ### Author Rebuttal · Reviewer_hSLS · 2026-04-01
> >
> > The authors’ rebuttal has addressed my primary concerns. Adding the promised discussions and clarifying the scope in the final version will better position this work in the research field.

---

> > > ### Author Response · Authors · 2026-04-03
> > >
> > > We are very glad to hear that our rebuttal addressed your primary concerns, and we sincerely appreciate your encouraging feedback. Thank you again for your time and thoughtful evaluation.

---

### Official Review · Reviewer_DJzA · 2026-03-11

**Soundness:** 4
**Presentation:** 4
**Significance:** 3
**Originality:** 4
**Overall Recommendation:** 5
**Confidence:** 4

**Summary:**

This paper discussed in RLVR the effect of the selection of tokens to training LLMs and the author discovery that the degree of off-policy is the key.
Specifically, the paper states under different degree of off-policy, the gradient will be dominate by different high-probability / low-probability tokens due to importance sampling clipping.

**Compliance With Llm Reviewing Policy:**

Affirmed.

**Final Justification:**

The paper is well-written and targets an important problem in RL4LLM.
Authors' rebuttal addresses most of my concerns.

**Key Questions For Authors:**

see Strengths And Weaknesses

**Strengths And Weaknesses:**

### Summary of Contributions

This paper investigates how the degree of off-policy data affects the clipping mechanism and subsequent parameter updates in Reinforcement Learning (RL). The authors address an interesting dichotomy in the field—namely, that training with low-probability tokens can lead to both exceptionally good or poor performance—and identify the degree of off-policy as a key differentiating factor. The authors provide robust theoretical analysis of the variance in the importance sampling (IS) ratio, supported by clear explanations and empirical validation. Overall, this is a well-motivated and clearly written paper whose strengths outweigh its weaknesses.

### Strengths

* **Strong Motivation and Clear Writing:** The paper effectively highlights the contradictory phenomena regarding low-probability training data. The subsequent investigation into the degree of off-policy provides a compelling and logical mechanism to explain these differences. The writing throughout the manuscript is highly fluent and accessible.
* **Robust Theoretical and Empirical Analysis:** The paper provides a rigorous theoretical examination of the variance of the importance sampling ratio, establishing it as a useful metric for the degree of off-policy. The mathematical propositions are accompanied by intuitive explanations, and the theoretical claims are well-validated by the empirical results presented in Figures 2 and 5.

### Weaknesses & Questions for the Authors

* **Reliance on PPO-Style Clipping and Token-Level IS:** The core logic of the paper relies heavily on a specific causal chain: *degree of off-policy $\rightarrow$ IS ratio $\rightarrow$ clipping $\rightarrow$ gradient $\rightarrow$ performance*. This raises two concerns regarding the generalizability of the findings:
    1.  **Relevance without Clipping:** Recent work, such as GPG [1], demonstrates that Reinforcement Learning with Verifiable Rewards (RLVR) can succeed without relying on IS ratios and clipping. Do the authors' analyses and conclusions still hold in these unclipped settings?
    2.  **Token-Level vs. Sequence-Level IS:** The analysis centers on token-level importance sampling. However, from a strict mathematical perspective, token-level IS is often considered flawed. Recent literature [2, 3] emphasizes the necessity of sequence-level importance sampling. It would strengthen the paper if the authors could clarify whether their analysis extends to sequence-level IS.
* **

[1] Chu, X., et al. "GPG: A Simple and Strong Reinforcement Learning Baseline for Model Reasoning."

[2] Pang, L., et al. "TIC-GRPO: Provable and Efficient Optimization for Reinforcement Learning from Human Feedback."

[3] Zheng, C., et al. "Group Sequence Policy Optimization."

[4] Zhu, H., et al. "The Path Not Taken: RLVR Provably Learns Off the Principals."

---

> ### Author Rebuttal · Authors · 2026-03-30
>
> Thank you for the thoughtful review and the strong endorsement of our work. We are glad that the reviewer found the motivation compelling and the theoretical analysis robust. We address the two generalizability questions below.
>
> > Q1 (W1): Relevance without clipping (e.g., GPG)
>
> A1: This is an important question about scope. Our contributions span two levels, and they differ in generality:
>
> - **Section 3 (token properties and advantage sign analysis):** These results characterize how token probability and advantage sign shape the policy gradient update. Since GPG's core is also a policy gradient method, this analysis applies regardless of whether clipping is used.
> - **Sections 2 and 4 (IS-ratio variance, clipping suppression, dominance reversal):** These results are specifically about the clipping mechanism and do not apply to unclipped settings like GPG.
> - **ACPO (Section 5):** Designed for clipped IS methods, which remain the dominant RLVR training paradigm.
>
> We view our work and clipping-free approaches like GPG as complementary: clipping remains important for training stability, but can introduce update distortion when applied uniformly; our analysis explains this mechanism and ACPO provides a principled fix within the clipped paradigm, while GPG explores an alternative path that removes clipping altogether. We will make this scope distinction more explicit in the revision.
>
> > Q2 (W1): Token-level vs. sequence-level IS
>
> A2: We agree this is a meaningful theoretical distinction. From a strict mathematical perspective, token-level IS has known limitations, as highlighted by the works the reviewer cites. Our analysis is indeed built on token-level IS, and in a sequence-level IS framework with sequence-level clipping (e.g., GSPO), the token dominance phenomenon we study would take a different form—the unit of clipping shifts from individual tokens to entire sequences, and the corresponding question becomes which sequences survive clipping rather than which tokens.
>
> That said, token-level IS remains the more widely adopted paradigm in current RLVR practice, and sequence-level methods like GSPO are particularly suited for scenarios with severe training-inference inconsistency (e.g., MoE models) rather than serving as a universal replacement. Our analysis characterizes the mechanism of the token-level paradigm that most existing methods operate in. Formally extending the analysis framework to sequence-level IS is an interesting future direction, and we thank the reviewer for the references.
>
> We would be glad to continue the discussion if any additional clarification would be helpful.

---

> > ### Author Rebuttal · Reviewer_DJzA · 2026-04-01
> >
> > The authors' rebuttal has addressed my concerns. I agree that exploring clipping-free RL algorithms for LLMs and sequence-level clipping falls outside the scope of this paper.
> >
> > However, I suggest modifying the title to avoid overclaiming. Adding a phrase such as 'An Analysis of Off-Policy Degree' or 'A Study from a Clipping Perspective' would better reflect the content and position the paper more accurately.

---

> > > ### Author Response · Authors · 2026-04-01
> > >
> > > Thank you very much for the thoughtful follow-up and for indicating that your concerns have been resolved. We also appreciate the suggestion on the title, and agree that the current phrasing can be made more precise to better reflect the paper's scope. In the final version, we will revise the title to more clearly emphasize the clipping-specific perspective and the role of off-policy degree.

---

### Official Review · Reviewer_En6d · 2026-03-15

**Soundness:** 2
**Presentation:** 2
**Significance:** 2
**Originality:** 3
**Overall Recommendation:** 4
**Confidence:** 3

**Summary:**

This paper studies why seemingly contradictory token-level heuristics in RLVR for LLM reasoning can all appear effective. The authors argue that the key hidden variable is the degree of off-policy training, which changes the dispersion of importance-sampling ratios and thus changes which tokens dominate the clipped policy-gradient update. Based on this analysis, they propose ACPO, which replaces a global clipping threshold with probability-bin-specific adaptive clipping thresholds, and show improvements over DAPO, CISPO, and entropy-based baselines on math, tabular QA, and arithmetic puzzle tasks.

**Compliance With Llm Reviewing Policy:**

Affirmed.

**Key Questions For Authors:**

See weaknesses

**Limitations:**

yes

**Strengths And Weaknesses:**

Strengths
- The proposed method, ACPO, is simple and easy to understand: it bins tokens by behavior-policy probability and adjusts clipping thresholds using within-bin IS-ratio dispersion. This makes the method relatively practical to implement on top of existing GRPO-style training.
- The empirical results covers multiple tasks and different model scales, both on-policy and off-policy settings show improvements.

Weaknesses
- The theoretical analysis relies on several strong assumptions whose empirical validity is not fully established. For example, the gaussian approximation for the IS ratio and the gaussian assumption on the advantage are convenient analytically, but it is not clear how accurate these assumptions are in real RLVR training.
- The reported improvements, while generally consistent, are not always large. On some settings the gains over strong baselines are modest.
- There are places where the presentation could be clearer. The paper introduces several levels of quantities: token-wise gradients, conditional expectations, grouped expectations, off-policy proxies, and subspace-angle analyses. Some readers may find it difficult to distinguish what is theoretical approximation, what is empirical observation, and what is the direct design principle used in ACPO.

---

> ### Author Rebuttal · Authors · 2026-03-30
>
> Thank you for the positive assessment and constructive feedback. We address each weakness below.
>
> > Q1 (W1): Theoretical assumptions
>
> A1: The Gaussian and independence assumptions are simplifications whose role is to yield Proposition 1's closed-form expression. Lemma 1, which states that lower-probability tokens have higher conditional IS-ratio variance, does not rely on them. Figure 2 further supports this structural trend empirically: lower-probability tokens exhibit larger IS-ratio spread and interact more unevenly with clipping. ACPO itself computes within-bin IS-ratio standard deviation directly from observed tokens, making no parametric assumption. If Gaussianity is relaxed, the current truncated-normal form is replaced by the corresponding truncated-moment expression for the chosen distribution; if independence is relaxed, the factorized form becomes a joint truncated expectation. These changes affect the compactness of the closed form, not the underlying mechanism. As long as lower-probability tokens have higher conditional IS-ratio variance and clipping suppression increases with this variance, the qualitative conclusion remains unchanged. In the camera-ready appendix, we will clarify this dependency structure and add discussion of how the expressions change under alternative distributional assumptions.
>
> > Q2 (W2): Modest gains
>
> A2:  Appendix K provides Welch's t-tests showing that ACPO significantly outperforms DAPO on multiple benchmarks, including Math500, AMC, AIME25, and Olympiad (p < 0.05). The absolute gains are also meaningful in several settings: on the ORZ/math task (7B), ACPO vs. DAPO improves Math500 off-policy by +3.76%, AMC off-policy by +4.36%, and Olympiad off-policy by +4.97%; on HiTab (3B, off-policy), the gain is +2.42%.
>
> AIME24 is a benchmark-specific exception where DAPO performs better. Our claim is therefore not that ACPO is uniformly best on every benchmark, but that it delivers consistent gains in the settings where fixed clipping causes the most distortion.
>
> > Q3 (W3): Presentation clarity
>
> A3: Thank you for pointing this out. The paper does introduce multiple analytical levels, and we agree the distinction between them could be made more explicit. To clarify the intended structure:
>
> - **Theoretical analysis** (Sections 2–3): Lemma 1 and Proposition 1 derive how clipping, advantage, and token probability jointly determine the effective update coefficient, leading to the gradient dominance reversal result. These are formal results under stated assumptions.
> - **Empirical observation** (Section 4.1, Figures): As visualized in Figure 2, the empirical behavior of IS-ratio spread and clipping differs systematically across probability bins, which motivates the method independently of the theory.
> - **Method design** (Section 4.2): ACPO's adaptive threshold $\epsilon_b = \epsilon_{base} + \alpha·\sigma_b$ is derived from the empirical observation, using within-bin IS-ratio standard deviation as the calibration signal.
>
> In the revision, we will add an explicit roadmap paragraph at the end of the introduction clarifying these three layers and how they connect.
>
> We also welcome further discussion if any additional clarification would be helpful.

---

> > ### Author Rebuttal · Reviewer_En6d · 2026-04-02
> >
> > The authors’ rebuttal has addressed my primary concerns and I'd like to maintain my positive score.

---

> > > ### Author Response · Authors · 2026-04-03
> > >
> > > We sincerely appreciate your positive assessment and are glad that our rebuttal addressed your primary concerns. If there are any follow-up questions or points that would benefit from further clarification, we would be very happy to continue the discussion, including via an updated Rebuttal Acknowledgement if that is the most convenient channel. Thank you again for your thoughtful evaluation.

---

### Decision · Program_Chairs · 2026-04-30

**Decision:**

Reject

**Comment:**

After reading the reviews, rebuttal, and the paper, I find the submission below the bar for a top-tier venue. Reviewers En6d, hSLS, and uYkU all agreed that the paper's central claim rests on **strong theoretical assumptions (Gaussian and independence) whose empirical validity has not been fully established**. Token-Level vs. Sequence-Level IS analysis lacks clarity (Reviewer DJzA). Concretely, I have carefully examined the authors' key assumption in the appendix: "assume the logit perturbation has variance independent of token probability." This assumption is not well justified. As the authors note, the policy gradient is influenced by interactions among a group of tokens. Consequently, it casts doubt on the core conclusions of the theoretical analysis, including Lemma 2.1 and Corollary 2.3. Regarding the experiments, all three reviewers (En6d, hSLS, and uYkU) initially found the improvements to be limited. The proposed ACPO seems to be an incremental refinement of DAPO rather than a major breakthrough. We encourage the authors to include more experiments and experimental details in the next version.